# Mechanical force regulates tendon extracellular matrix organization and tenocyte morphogenesis through TGFbeta signaling

Arul Subramanian[1], Lauren Fallon Kanzaki[1], Jenna Lauren Galloway[2], Thomas Friedrich Schilling[1]*

[1]Department of Developmental and Cell Biology, University of California, Irvine, Irvine, United States; [2]Center for Regenerative Medicine, Department of Orthopaedic Surgery, Massachusetts General Hospital, Harvard Stem Cell Institute, Boston, United States

**Abstract** Mechanical forces between cells and extracellular matrix (ECM) influence cell shape and function. Tendons are ECM-rich tissues connecting muscles with bones that bear extreme tensional force. Analysis of transgenic zebrafish expressing mCherry driven by the tendon determinant *scleraxis* reveals that tendon fibroblasts (tenocytes) extend arrays of microtubule-rich projections at the onset of muscle contraction. In the trunk, these form a dense curtain along the myotendinous junctions at somite boundaries, perpendicular to myofibers, suggesting a role as force sensors to control ECM production and tendon strength. Paralysis or destabilization of microtubules reduces projection length and surrounding ECM, both of which are rescued by muscle stimulation. Paralysis also reduces SMAD3 phosphorylation in tenocytes and chemical inhibition of TGFβ signaling shortens tenocyte projections. These results suggest that TGFβ, released in response to force, acts on tenocytes to alter their morphology and ECM production, revealing a feedback mechanism by which tendons adapt to tension.
DOI: https://doi.org/10.7554/eLife.38069.001

*For correspondence:
tschilli@uci.edu

Competing interests: The authors declare that no competing interests exist.

## Introduction

Cells in all multicellular organisms are exposed to mechanical forces through adhesions to neighboring cells and to the extracellular matrix (ECM), as well as the ebb and flow of the environment. Force has been shown to influence cellular processes such as cell division, survival, migration, and differentiation (*Behrndt et al., 2012*; *Culver and Dickinson, 2010*; *Hamada, 2015*; *Keller et al., 2008*; *Roman and Pekkan, 2012*). Cellular responses to force include the activation of cell surface receptors such as integrins (Itgs), G-protein-coupled receptors (GPCRs), transient receptor potential (TRP) ion channels, and Piezo channels (*Busch et al., 2017*; *Chachisvilis et al., 2006*; *Maartens and Brown, 2015*; *Mederos y Schnitzler et al., 2008*; *Popov et al., 2015*; *Wu et al., 2017*). Despite recent insights into the nature of such responses, few in vivo studies have investigated how cells adapt to force and alter the ECM landscape to strengthen or weaken it accordingly (*Maeda et al., 2011*; *Ng et al., 2014*).

The musculoskeletal system bears among the strongest forces experienced by any tissue, such as the tensional forces exerted upon tendons and ligaments (*Heinemeier et al., 2013*; *Wang, 2006*). Tendons can withstand such forces due to the specialized organization of collagen (Col) fibers and proteoglycans within each tendon fibril. Tendon injuries are extremely common and debilitating, especially in athletes, the elderly, and patients with neuromuscular diseases such as muscular

**eLife digest** Tendons – the fibrous structures that attach muscles to bones – must withstand some of the strongest forces in the body. Little is known about how tendons develop or adapt to withstand these forces. Studies have shown that muscles respond actively to force, as seen during exercise. Do tendons respond in similar ways?

Tendons consist of collagen fibers surrounded by a 'matrix' of proteins. Also embedded in the matrix are specialized cells called tenocytes, which regulate the production of the different components of the tendon. A genetic modification allows tenocytes to be tracked using a fluorescent gene product that can be viewed using a microscope. Subramanian et al. have now used this technique in zebrafish to watch how the behaviors of the tenocytes change in response to forces applied to the tendon.

Subramanian et al. show that at the start of muscle contraction, tenocytes put forth long projections from their cell bodies that extend perpendicular to the muscle fibers. This suggests that the projections act as force sensors. Consistent with this idea, paralyzing the muscle causes the projections to shrink. This shrinkage correlates with changes in how the tendon matrix proteins are organized.

Further investigation reveals a force-responsive signaling pathway in the tenocytes that controls how these cells grow and produce key tendon matrix proteins. Subramanian et al. believe this pathway is central to how tendons adapt to the forces applied during muscle contraction.

A better knowledge of how force affects tendon structure could ultimately help to improve treatments for tendon injuries and tendon atrophy. In particular, understanding how force affects how tenocytes develop could help researchers to develop new ways to regenerate and repair tendons.

DOI: https://doi.org/10.7554/eLife.38069.002

dystrophy (*Bönnemann, 2011*; *Walden et al., 2017*). Despite their prevalence, little is known about how tendon fibroblasts (tenocytes) respond in vivo to tensional force at muscle attachments, or how they adapt to changes in mechanical load. Tendons form a variety of attachment sites - connecting muscles to cartilages and bones as well as other muscles and soft tissues. A myotendinous junction (MTJ) is a specialized ECM-rich region at the interface of muscle-tendon attachment sites that functions as the primary sources of force transmission. Each type of attachment bears varying levels of force, which correlates with distinct composition and organization of its tendon ECM (*Ker et al., 2000*; *Wang, 2006*). While extensive research has been conducted to evaluate the effects of exercise on size and strength of muscle fibers, less is known about how it effects tendon morphology and function. Understanding this is key to gaining insights into the causes of tendon defects and developing new treatments for tendon injuries or atrophy.

Previous studies in vitro have suggested that tenocytes actively respond to changes in force in their environment by modulating ECM composition and organization (*Maeda et al., 2010*; *Rullman et al., 2009*). Excised tendons stretched in collagen gels, as well as tissue samples from chronic Achilles tendonitis patients, upregulate various collagens and ECM-modulating proteins, particularly Col3, Matrix Metalloproteinase 9 (MMP9) and MMP13 (*Ireland et al., 2001*; *Pingel et al., 2014*). In addition, collagen fibril size decreases and fibril packing increases in tendinopathies, likely due to increased ECM turnover (*Pingel et al., 2014*). These studies have suggested ECM modifications and morphological changes in tendinopathies but they have largely been limited to cultured tendons or tendon fragments. In vivo, several growth factor signaling pathways and transcription factors have been implicated downstream of mechanical force in tendon development and repair in mice. These include several members of the Transforming Growth Factor (TGF) superfamily, including TGFβ and Bone Morphogenetic Proteins (BMPs), as well as Fibroblast Growth Factors (FGF) (*Gumucio et al., 2015*; *Nourissat et al., 2015*). Mice lacking the transcription factor Scleraxis (Scx) show severe defects in force-transmitting and load-bearing tendons, suggesting that Scx is essential for maintenance of tendon ECM in response to mechanical force (*Murchison et al., 2007*). In addition, Scx directly regulates transcription of tendon ECM components, including Col1a (*Havis et al., 2014*; *Subramanian and Schilling, 2015*). Our studies of the ECM protein Thrombospondin 4b

(Tsp4b) in zebrafish have shown that it is an essential scaffolding protein for tendon ECM assembly, required to maintain muscle attachments subjected to mechanical force via muscle contraction, and able to strengthen attachments when overexpressed (*Subramanian and Schilling, 2014*).

Here, we show that mechanical force causes remarkable morphological changes in tenocytes in zebrafish, which form a dense curtain of projections at MTJs, and in their surrounding ECM. Tenocyte projections have been reported in electron micrographs of mammalian tendon fascicles yet how they form and their functions in tendon development remain largely unexplored (*Kalson et al., 2015*; *Knudsen et al., 2015*). Our results suggest that tenocytes play an active role in sensing force and thereby regulating ECM composition and overall tendon strength. In addition, we show that the force of muscle contraction regulates the growth and branching of tenocyte projections via TGFβ signaling. Such feedback between tenocytes and ECM may be a common mechanism for force adaptation within the musculoskeletal system.

## Results

### Tenocytes elongate with the onset of muscle contraction

Tenocytes in zebrafish express two *Scx* orthologues, *scxa* and *scxb* (*Chen and Galloway, 2014*). Using a bacterial artificial chromosome (BAC) transgenic line that expresses *mCherry* under the control of regulatory elements for *scxa*, Tg(*scxa:mCherry*), we examined the morphogenesis of tenocytes during embryonic (20 hr post fertilization (hpf) to 72 hpf) and early larval (72 hpf to 5 dpf) zebrafish development. Expression of *scxa:mCherry* was first detected at 20 hpf in muscle and tendon progenitors of the somites. In a developing zebrafish embryo, muscles in the trunk establish attachments at bilateral, 'chevron' shaped somite boundaries that subdivide each muscle segment forming the vertical myoseptum (VMS). In addition, dorsal and ventral compartments within each somite are subdivided by a horizontal myoseptum (HMS), which extends laterally from the notochord (NC), along which oblique myofibers attach. By 24 hpf, as the first myofibers differentiated, *scxa:mCherry* expression in muscle progenitors diminished and became progressively restricted to scattered tendon progenitors along the HMS and VMS (~24 cells per VMS) (*Figure 1A, D,G*) (*Figure 1—video 1*). Cells with the highest levels of *scxa:mCherry* expression were located laterally, adjacent to the HMS, while more medial cells expressed lower levels (*Figure 1A', D', G'*). By 36 hpf, *scxa:mCherry+* cells doubled in number (~44/VMS) and became increasingly localized to the HMS and VMS at future MTJs (*Figure 1B,E,H*) (*Figure 1—video 1*). At this stage, cells with the highest *scxa:mCherry* expression that were located medially and in the ventral somites began to extend projections laterally along the VMS, perpendicular to the orientation of muscle fibers (*Figure 1B', E', H'*). By 48 hpf these projections extended 70–80 µm (*Figure 1C,F,I*). 3D-reconstructions of confocal stacks at 60 hpf revealed that this polarized network of tenocyte projections covered the entire VMS (*Figure 1C', F', I'*; *Figure 1—figure supplement 1*). Time-lapsed videos of Tg(*scxa:mCherry*) embryos capturing images at 20 min intervals from 48 to 60 hpf showed that tenocyte projections are dynamic and constantly changing in length and branching pattern (*Figure 1—video 2*). Tenocytes along the HMS near the NC have shorter, more convoluted projections than tenocytes along the VMS (*Figure 1I'*; *Figure 1—figure supplement 1B*). Thus, during the period in which axial muscles in the trunk begin to contract and embryos become motile, tenocytes align along future MTJs and undergo dramatic changes in cell shape that correlate with the establishment and strengthening of muscle attachments.

Cranial tendons also undergo dramatic morphological changes during the onset of muscle attachment and contractility. A cluster of *scxa:mCherry+* tenocyte progenitors is first observed at 36 hpf in the ventral midline near the future attachment sites of the sternohyoideus (SH) and adductor mandibulae (AM), which are among the earliest muscles to differentiate at 53 hpf (*Schilling and Kimmel, 1997*). By 48 hpf, three major clusters of *scxa:mCherry+* tenocytes are visible ventrally, one anterior that forms ventral mandibular, hyoid, and oculomotor muscle tendons and two posterior clusters associated with each SH (*Figure 1—figure supplement 2A*). The anterior cluster subdivides over 14 hr into separate attachment sites for mandibular (IMA, IMP, AM) and hyoid (IH) muscles (*Figure 1—figure supplement 2B*). Double labeling with anti-MHC and anti-mCherry antibodies revealed a tight correlation between the timing of the onset of muscle contraction and tenocyte morphogenesis in each of these clusters (*Figure 1—figure supplement 2A,B,D,E,G,H*). Cranial myofibers remain

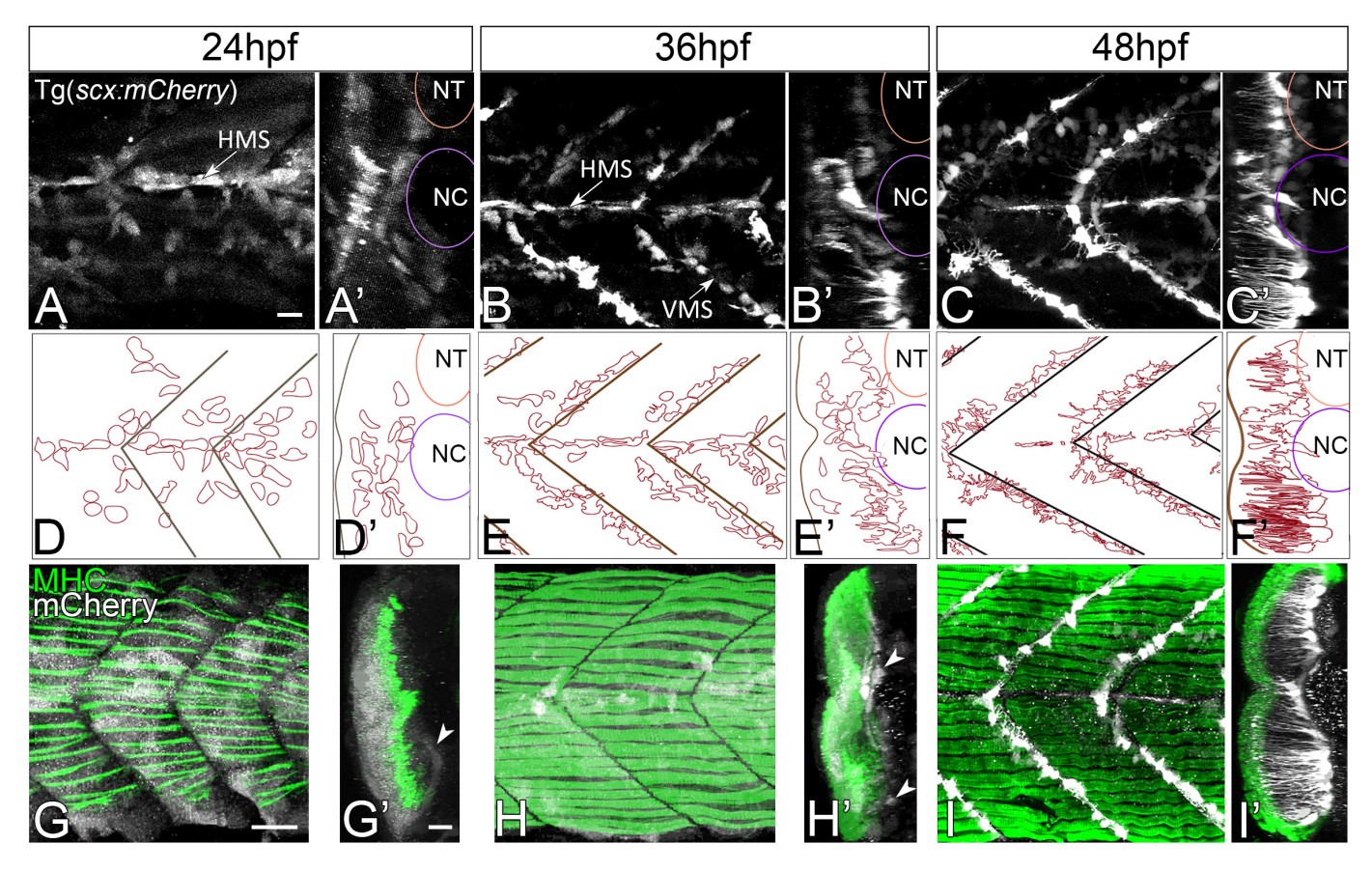

**Figure 1.** Axial tenocyte morphogenesis. (**A–C**) Lateral views of live Tg(*scx:mCherry*) embryos showing developing tenocytes (A - 24 hpf, B - 36 hpf, C - 48 hpf). (**A'–C'**) Transverse views from 3D projections showing the positions of developing tenocytes in relation to the notochord (NC) and neural tube (NT) along the horizontal (HMS) and vertical myosepta (VMS) (arrows). Tenocytes form projections at 36–48 hpf (**B' and C'**). (**D–F**) Diagrams of lateral views showing the morphology of tenocytes in the developing somites. (**D'–F'**) Diagrams of transverse views from 3D projections of live Tg(*scx:mCherry*) embryos show the development of tenocyte projections (**E' and F'**). (**G–I**) Lateral views of co-immunostained Tg(*scx:mCherry*) embryos showing developing tenocytes (anti-mCherry - white) and muscle fibers (anti-MHC - green) (G – 24 hpf, H – 36 hpf, I – 48 hpf). (**G'–I'**) Transverse views from 3D projections of live Tg(*scx:mCherry*) embryos showing the positions of developing tenocytes (arrowheads in G' and H') in relation to the myotome. Scale bars = 20 microns.

DOI: https://doi.org/10.7554/eLife.38069.003

The following video and figure supplements are available for figure 1:

**Figure supplement 1.** Axial tenocytes form polarized projections orthogonal to muscle fibers.

DOI: https://doi.org/10.7554/eLife.38069.004

**Figure supplement 2.** Cranial tenocyte morphogenesis correlates with onset of muscle contraction.

DOI: https://doi.org/10.7554/eLife.38069.005

**Figure 1–video 1.** Axial tenocyte progenitors align along HMS and VMS following muscle fiber differentiation.

DOI: https://doi.org/10.7554/eLife.38069.006

**Figure 1–video 2.** Tenocyte projections are dynamic.

DOI: https://doi.org/10.7554/eLife.38069.007

immotile at 48–62 hpf and their corresponding tenocytes form clusters of rounded cells at future attachment sites. These tenocytes then undergo compaction and elongation as contractions begin at 72 hpf (*Figure 1—figure supplement 2C,F,I*).

## Tenocyte elongation requires muscle contraction

Based on the close correlation between the onset of muscle contraction and tenocyte morphogenesis, we hypothesized that mechanical force serves as a cue for tenocytes to elongate and form

projections. To test this idea, we first injected full-length mRNA encoding codon-optimized $\alpha$-bungarotoxin ($\alpha$Btx), a specific irreversible antagonist of acetylcholine receptors that blocks neuromuscular synapses and prevents skeletal muscle contractions (*Swinburne et al., 2015*; *Westerfield et al., 1990*). Embryos injected with $\alpha$Btx mRNA at the one-cell stage were completely paralyzed until 60 hpf, after which they gradually recovered motility as $\alpha$Btx activity declined. Depth-coded, 3D-reconstructed images of living trunk tenocytes along the MTJs of somites 16–17 at 48 hpf revealed an average reduction of 13 μm (18%) in axial tenocyte projection length in $\alpha$Btx-injected embryos compared to uninjected controls (*Figure 2A,B,E*). Paralyzed embryos also showed reduced branching complexity in their projections (*Figure 2F*) and projection density along the VMS (*Figure 2—figure supplement 1*). To restore mechanical force, we electrically stimulated $\alpha$Btx-injected embryos to induce muscle contractions, as described previously (*Subramanian and Schilling, 2014*). Stimulation at 48 hpf for 2 min at 20V caused no visible muscle damage or significant change in tenocyte projection lengths compared to controls (*Figure 2C,E*) while the same stimulation of $\alpha$Btx-injected embryos rescued both tenocyte projection length and density along the VMS almost completely (*Figure 2D,E*; *Figure 2—figure supplement 1*). The observed reductions in projection length and density were caused by paralysis rather than any unanticipated effect of $\alpha$Btx, since homozygous mutants paralyzed due to lack of a functional voltage-dependent L-type calcium channel subtype beta-1 (Cacnb1), necessary for excitation-contraction coupling in muscle, showed similar (10–15 μm) reductions in projection length (*Figure 2—figure supplement 2*) (*Zhou et al., 2006*). Tenocytes in *cacnb1* mutant embryos fail to compact and elongate. Since $\alpha$Btx-injected embryos recover from paralysis at 65 hpf, prior to cranial muscle contractions, we compared cranial tenocyte patterning in immunostained 4 dpf *cacnb1* mutant embryos with their siblings. We observed both a failure of cranial tenocytes to compact and elongate, as well as reduced projections and frayed myofibers (*Figure 2—figure supplement 3*). These results indicate a strong correlation between mechanical force from muscle contraction and tenocyte morphogenesis, suggesting that force stimulates the dynamic growth and branching of tenocyte projections.

## ECM organization at MTJs requires muscle contraction

We previously showed that Tsp4b secreted by tenocytes is essential for ECM organization at MTJs and strengthens muscle attachments (*Subramanian and Schilling, 2014*). We hypothesized that force stimulates tenocytes to secrete Tsp4b from the projections they extend into the tendon ECM. Consistent with this, injection of *tsp4b-gfp* full length mRNA into Tg(*scxa:mCherry*) embryos produced Tsp4b-GFP protein that localized to MTJs along the attachment sites at 48 hpf (*Figure 3A–C, I*). This exogenous Tsp4b-GFP protein was dramatically reduced in $\alpha$Btx-injected embryos, particularly around projections, and became diffuse compared to uninjected controls (*Figure 3D–F,I*). Likewise, immunohistochemical staining for Tsp4b at 48 hpf in $\alpha$Btx-injected embryos showed dramatic reductions along the attachment sites compared to controls (*Figure 3G,H,J*). In contrast, other ECM proteins such as laminin (Lam) at 48 hpf and and fibronectin (Fn) at 24 hpf showed no significant changes at the MTJ in $\alpha$Btx-injected embryos at 48 hpf (*Figure 3—figure supplement 1*). Defects in Tsp4b distribution were due to the lack of mechanical force, since restoring force in paralyzed embryos through electrical stimulation rescued both local levels and the overall area of Tsp4b protein localization along the VMS (*Figure 3—figure supplement 2*). To test the hypothesis that changes in Tsp4b localization were due to reduced *tsp4b* gene expression in response to lack of force, we performed real-time PCR and found a significant reduction in *tsp4b* expression at 48 hpf in $\alpha$Btx-injected embryos, while no significant change in expression was observed at 24 hpf (*Figure 3—figure supplement 3*). These results suggest a role for mechanical force in both assembly of tendon ECM and expression of key MTJ ECM genes during development and demonstrate that muscle contractions regulate the composition and organization of the tendon ECM.

## Microtubules maintain tenocyte projections and their interactions with tendon ECM

Cellular projections in neurons, keratinocytes and pigment cells are rich in microtubules (MTs) and in some cases F-actin, while filopodial extensions of cells are typically more actin-based (*Eom et al., 2015*; *Witte et al., 2008*). To determine the cytoskeletal structure of tenocyte projections we injected full-length mRNA encoding *eGFP-αtubulin* and found that this fusion protein localized to

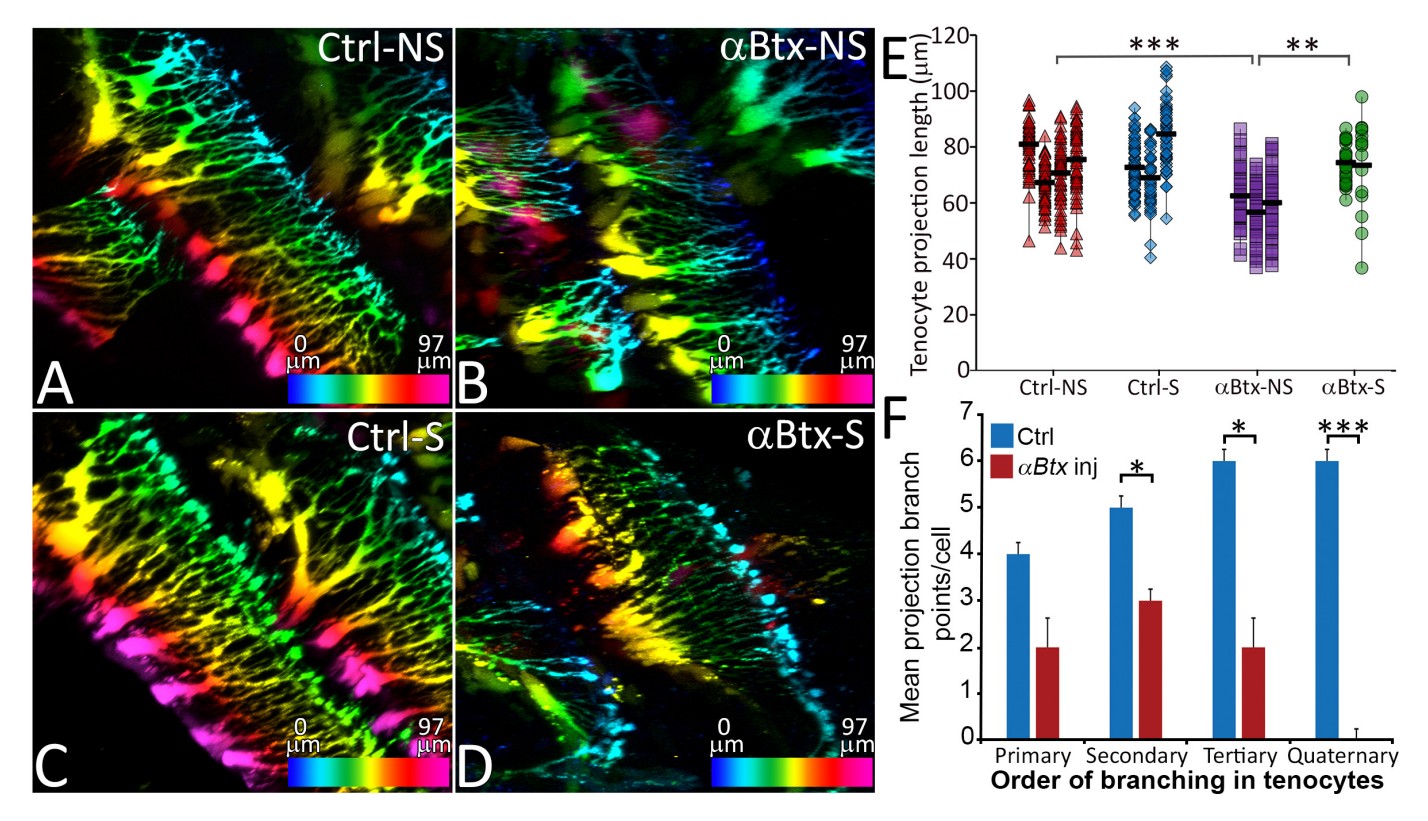

**Figure 2.** Tenocyte projection length and branching density is regulated by mechanical force. Lateral views of live Tg(*scx:mCherry*) embryos (48 hpf) showing tenocyte projections. Images are pseudocolored by depth from medial (red) to lateral (blue). Control embryos were imaged without stimulation (**A**) and after stimulation (**B**), and the length of tenocyte projections was compared with embryos injected with αBtx and imaged without (**C**) and with stimulation (**D**). Dot plot shows individual data points of tenocyte projection length under different conditions (**E**). The data points from each embryo are connected by a vertical line. NS – Not Stimulated, S – Stimulated. (n > 50 data points/embryo in three embryos/sample, p value was determined through ANOVA 1-way analysis ***<0.00001, **<0.0001). Histogram shows quantification of branch points along tenocyte projections per tenocyte in 36 hpf control and αBtx injected embryos for every level of branching (1° – primary, 2° – secondary, 3° – tertiary, 4° – quaternary). (n = 4, p value was determined through ttest *<0.01, ***<0.00001). The measurements used for quantitative analysis and creation of the plots can be accessed from *Figure 2—source data 1* and *Figure 2—source data 2*.

DOI: https://doi.org/10.7554/eLife.38069.008

The following source data and figure supplements are available for figure 2:

**Source data 1.** Measurements of tenocyte projection length along VMS.
DOI: https://doi.org/10.7554/eLife.38069.014

**Source data 2.** Measurement of tenocyte projection branching complexity along VMS.
DOI: https://doi.org/10.7554/eLife.38069.015

**Figure supplement 1.** Density of tenocyte projections is regulated by mechanical force.
DOI: https://doi.org/10.7554/eLife.38069.009

**Figure supplement 1—source data 1.** Measurements of projection density along VMS.
DOI: https://doi.org/10.7554/eLife.38069.010

**Figure supplement 2.** *cacnb1* mutants show reduced length and branching of tenocyte projections.
DOI: https://doi.org/10.7554/eLife.38069.011

**Figure supplement 2—source data 1.** Measurements of Tsp4b localization area.
DOI: https://doi.org/10.7554/eLife.38069.012

**Figure supplement 3.** Cranial tenocyte patterning and morphogenesis is disrupted in *pet* mutants.
DOI: https://doi.org/10.7554/eLife.38069.013

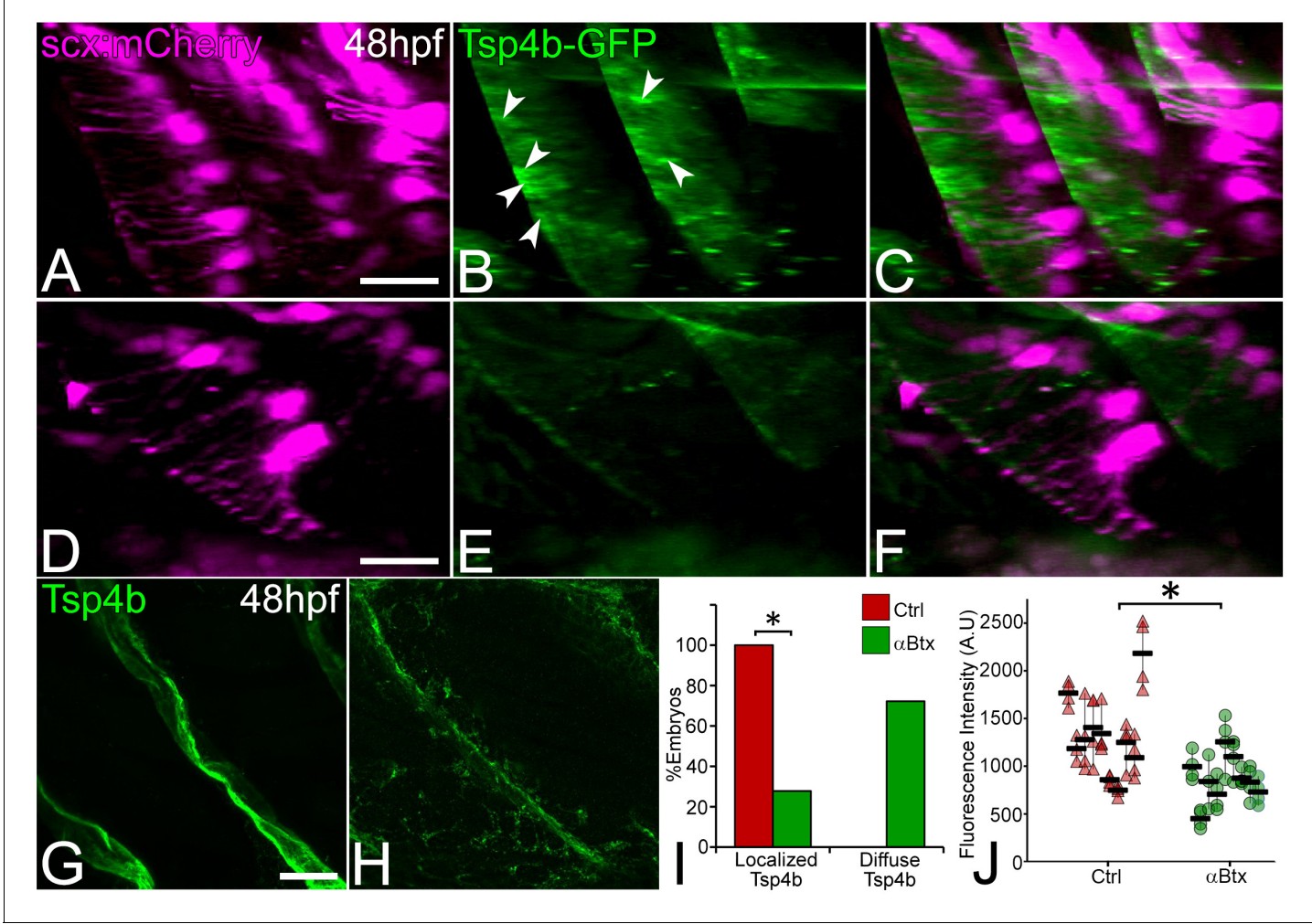

**Figure 3.** Tsp4b localization to VMS and tenocyte projections requires mechanical force. Lateral views of live control (**A–C**) and αBtx injected (**D–F**) Tg (*scx:mCherry*) embryos (48 hpf), injected with *tsp4b-gfp* mRNA showing localization of Tsp4b-GFP (green) (arrowheads) along the VMS and tenocyte projections (red). (**I**) Histogram shows the percentage of embryos with Tsp4b-GFP localized to VMS (n = 27, p value calculated by chi-squared test <0.05). (**G–H**) Lateral views of immunostained embryos showing Tsp4b protein localization detected immunohistochemically along VMS in control (**G**) and αBtx injected (**H**) embryos. (**J**) Dot plot shows individual data points of the fluorescent intensity of localized Tsp4b along the VMS in control and αBtx injected embryos. Three VMSs/embryo were sampled in control and αBtx-injected embryos. (n = 9, p value calculated by Wilcoxon Rank Sum Test - < 0.0001). Scale bars = 20 microns. The measurements used for quantitative analysis and creation of the plots can be accessed from *Figure 3—source data 1* and *Figure 3—source data 2*.

DOI: https://doi.org/10.7554/eLife.38069.016

The following source data and figure supplements are available for figure 3:

**Source data 1.** Count of embryos showing localized or diffuse Tsp4b-GFP.
DOI: https://doi.org/10.7554/eLife.38069.023

**Source data 2.** Measurements of Tsp4b fluorescence intensities along VMS.
DOI: https://doi.org/10.7554/eLife.38069.024

**Figure supplement 1.** Early Lam and Fn organization do not depend on mechanical force.
DOI: https://doi.org/10.7554/eLife.38069.017

**Figure supplement 1—source data 1.** Measurement of Laminin fluoresence intensity along VMS.
DOI: https://doi.org/10.7554/eLife.38069.018

**Figure supplement 1—source data 2.** Measurement of Fibronectin fluoresence intensity along VMS.
DOI: https://doi.org/10.7554/eLife.38069.019

**Figure supplement 2.** Tsp4b organization requires mechanical force.
DOI: https://doi.org/10.7554/eLife.38069.020

**Figure supplement 2—source data 1.** Measurements of Tsp4b localization area.

*Figure 3 continued on next page*

*Figure 3 continued*

DOI: https://doi.org/10.7554/eLife.38069.021

**Figure supplement 3.** Mechanical force regulates expression of Tsp4b.

DOI: https://doi.org/10.7554/eLife.38069.022

MTs along the length of tenocyte projections        (*Figure 4A–C*) (*Rusan et al., 2001*). Similar injections of plasmids encoding *EGFP-Lifeact-7* failed to show labeled actin in the projections. To determine if MTs are critical for maintaining projections, we treated embryos with Nocodazole, which caused them to fragment (*Figure 4D,E*). Immunohistochemical staining of Nocodazole-treated embryos for Tsp4b showed scattered Tsp4b + puncta localized at MTJs along the VMS and reduced Tsp4b protein levels in the VMS (*Figure 4F,G,H,I*). These results suggest that MTs are the key structural components of tenocyte projections required to sustain the organization of tendon ECM.

## TGFβ signaling is required for tenocytes to extend projections in response to force

Previous studies from primary cultures of tenocytes and stretch tests on isolated tendons in vitro have proposed a mechanoresponsive role for TGFβ signaling (*Gumucio et al., 2015*; *Havis et al., 2016*; *Maeda et al., 2011*). TGFβ secreted by muscles or latent in the ECM of the MTJ could be released in response to force and thereby regulate both tenocyte morphogenesis and ECM production. To address this hypothesis, we treated Tg(*scxa:mCherry*) embryos with a chemical inhibitor of TGFβ signaling (SB431542 – which blocks TGFβ receptors) for 12 hr from 24 to 36 hpf (*Chen and Galloway, 2014*). This treatment severely reduced signaling in both muscle fibers and tenocytes as confirmed by immunostaining for phosphorylated SMAD3 (pSMAD3) in SB431542-treated embryos compared to controls (*Figure 5A–C,E–G,I*). In addition, tenocyte projections were reduced in length by an average of ~20 μm in SB431542-treated embryos (*Figure 5D,H,J*), similar to the effects of αBtx (*Figure 2B,E*). However, unlike embryos injected with αBtx, SB431542-treated embryos continued to swim actively. These results suggest that TGFβsignaling acts downstream of muscle contraction to stimulate growth and branching of tenocyte projections. To confirm if muscle contraction is essential for activation of TGFβ signaling, we stained control and αBtx-injected, Tg(*scx:mCherry*) embryos with anti-pSMAD3. While control embryos showed strong pSMAD3 localization in the nuclei of muscles and tenocytes, pSMAD3 staining was strongly reduced in the nuclei of tenocytes in αBtx-injected embryos (*Figure 6A–G*). Here, in contrast to embryos treated with SB431542 (*Figure 5C,G*), paralysis specifically reduced pSMAD3 in tenocytes and not in muscle nuclei. This correlated with the reduction in length of tenocyte projections (*Figure 6H*). These results suggest that mechanical force from muscle contraction serves as a cue for TGFβ mediated signaling in tenocytes to control their morphogenesis and differentiation.

To further confirm that mechanical force has a role in induction of TGF-β responses in tenocytes, we stained control and αBtx-injected embryos with or without electrical stimulation, with anti-pSMAD3 antibody to verify if localization of pSMAD3 in nuclei of tenocytes could be rescued. αBtx-injected embryos stimulated with mild electric current showed increased pSMAD3 localization in tenocyte nuclei strongly suggesting that mechanical force from muscle contraction can rescue TGFβ signaling in tenocytes (*Figure 6—figure supplement 1*).

## Tenocyte projections regulate force-dependent gene expression

Previous studies have linked mechanical force with the expression of tenogenic and myogenic genes (*Chen et al., 2012*; *Maeda et al., 2011*). Our results showing similar tenocyte projection defects in Nocodazole-treated, αBtx-injected and SB431542 treated embryos suggest that they induce similar changes in expression of force-responsive genes. Real-time PCR analysis on cDNA prepared from 48 hpf control and αBtx-injected embryos revealed that paralysis led to an almost complete loss of expression of *tsp4b*, as well as *TGFβ-induced protein* (*tgfbip*) and, *connective tissue growth factor a* (*ctgfa*), while expression levels of other tendon genes, such as *scxa*, were unaffected (*Figure 6—figure supplement 2A*). All three genes (*tsp4b, tgfbip* and *ctgfa*) were restored to control levels of expression with electrical stimulation (*Figure 6—figure supplement 2A*). We further validated the results with digital droplet PCR (ddPCR) on cDNA prepared from FACS sorted tenocytes and whole

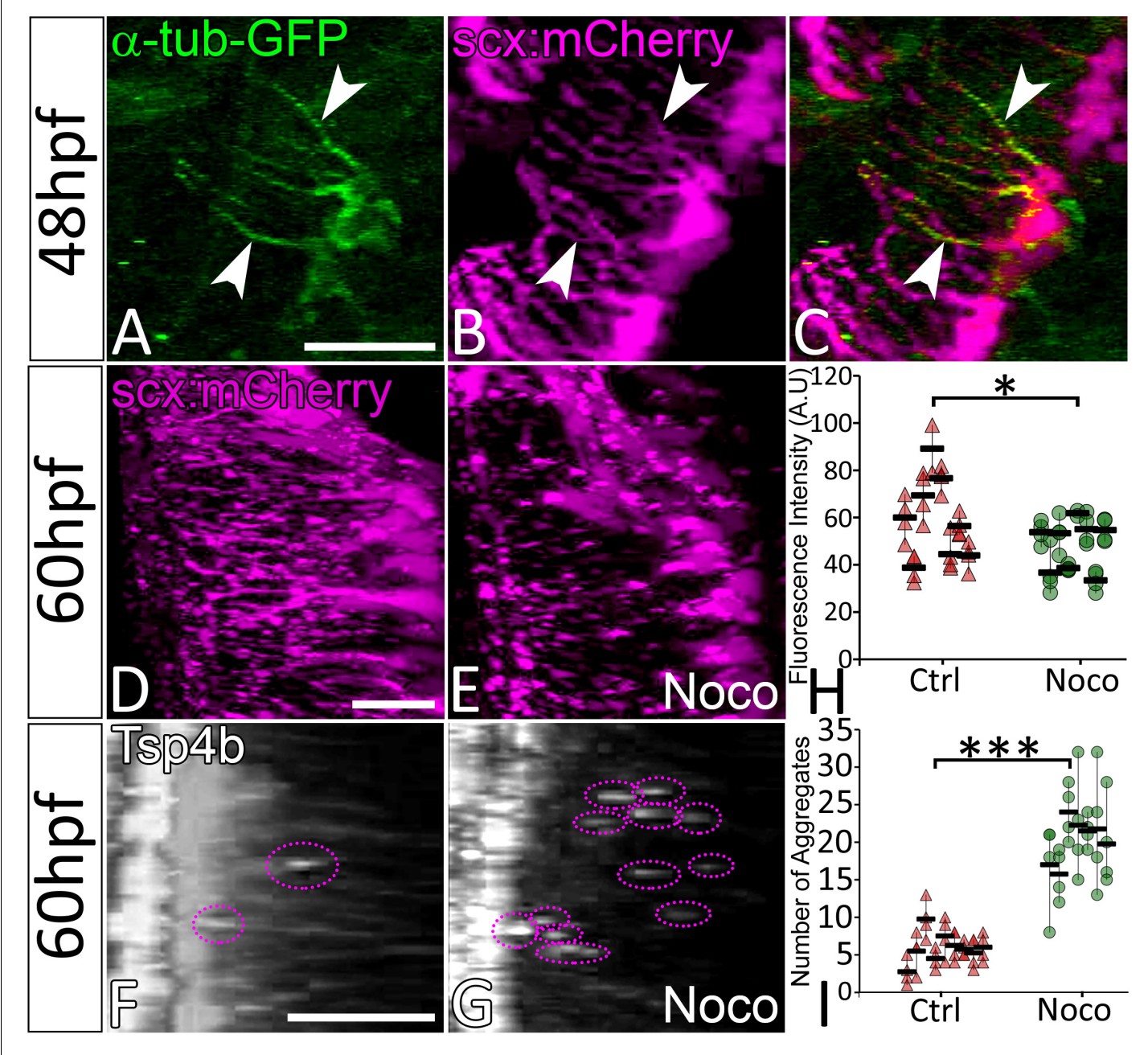

**Figure 4.** Microtubule-rich tenocyte projections control tendon ECM localization. Lateral views of live 48 hpf Tg(*scx:mCherry*) embryos injected with EGFP-alpha-Tubulin mRNA (**A–C**) showing localization of a-Tubulin along the length of projections colocalized with mCherry to mark in tenocytes. Transverse views of 3-D reconstructed live 60 hpf embryos showing tenocyte projections in DMSO-treated (**D**) and Nocodazole (Noco)-treated (**E**) embryos. Transverse view of 3-D reconstructed 60 hpf embryos immunostained for Tsp4b showing localization of Tsp4b in DMSO treated (**F**) and Noco treated (**G**) samples. Quantification of Tsp4b localization intensity in VMS (**H**) and distribution of Tsp4b aggregates in VMS (**I**) of DMSO-treated and Noco-treated embryos. (p value calculated by t-test for samples with unequal variance *<0.05, ***<0.0005). Scale bars = 20 microns. The measurements used for quantitative analysis and creation of the plots can be accessed from *Figure 4—source data 1* and *Figure 4—source data 2*.
DOI: https://doi.org/10.7554/eLife.38069.025

The following source data is available for figure 4:

**Source data 1.** Mesurements of Tsp4b fluorescence intensities along VMS.
DOI: https://doi.org/10.7554/eLife.38069.026
**Source data 2.** Count of Tsp4b aggregates along VMS.
DOI: https://doi.org/10.7554/eLife.38069.027

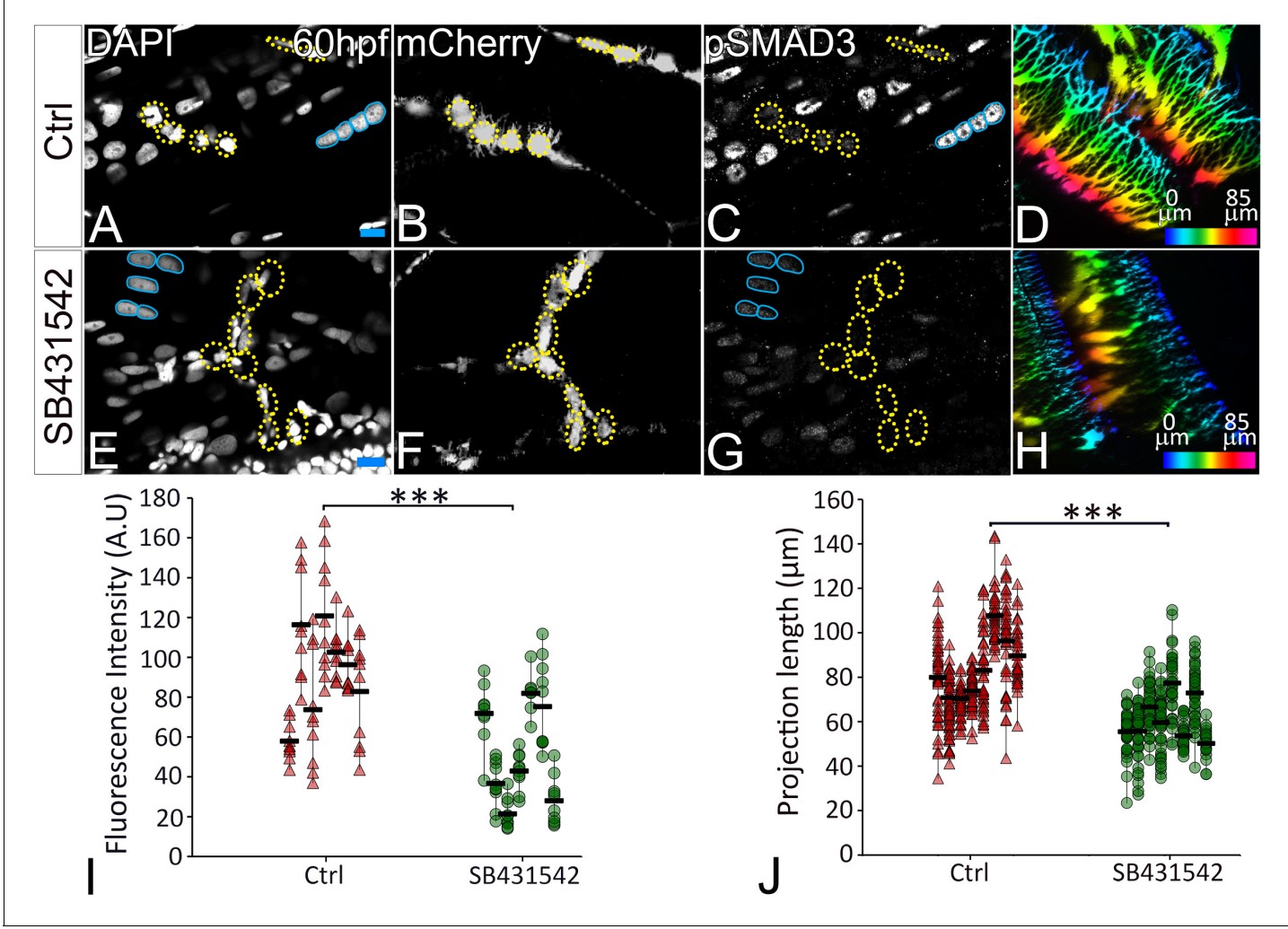

**Figure 5.** TGFβ signaling regulates tenocyte morphogenesis. Lateral views of immunostained Tg(*scx:mCherry*) control (**A–D**) and SB431542-treated (**E–H**) embryos showing nuclei (DAPI), tenocytes (anti-mCherry) and pSMAD3 (anti-pSMAD3). (**I**) Localization of pSMAD3 was quantified as fluorescent intensity of nuclear pSMAD3 signal (marked by yellow dotted ROI) and plotted as a dot plot showing data points (n = 9, p value was calculated by t test ***<0.000005). (**D, H**) Pseudocolored 3D projections show tenocyte cell projections in control (**D**) and SB 431542 treated embryos (**H**). (**J**) Dot plot shows individual data points representing tenocyte projection lengths (n = 50 data points/embryo in nine embryos/sample, p value was calculated by Wilcoxon Rank Sum test ***<0.00005). Representative muscle nuclei are marked by a blue continuous ROI. Scale bars = 10 microns. The measurements used for quantitative analysis and creation of the plots can be accessed from *Figure 5—source data 1* and *Figure 5—source data 2*.

DOI: https://doi.org/10.7554/eLife.38069.028

The following source data is available for figure 5:

**Source data 1.** Measurements of pSMAD3 fluorescence intensities in tenocyte nuclei along VMS.
DOI: https://doi.org/10.7554/eLife.38069.029
**Source data 2.** Measurements of tenocyte projection length along VMS.
DOI: https://doi.org/10.7554/eLife.38069.030

embryos respectively. Our ddPCR results from whole embryo cDNA preparation agreed with the real-time PCR analysis (*Figure 6—figure supplement 2B*). Real-time PCR analysis of SB431542-treated embryos showed significant reductions in expression of *tsp4b* and *tgfbip* genes (*Figure 6—figure supplement 3A*). Loss of tenocyte projections through destabilization of microtubules in nocodazole-treated embryos also led to reduced expression of *tsp4b* and *tgfbip* while expression of *ctgfa* and *scxa* were elevated (*Figure 6—figure supplement 3B*). Taken together, these results are consistent with the hypothesis that mechanical force acts through TGFβ signaling to regulate teno-cyte-specific transcription including ECM components such as Tsp4b.

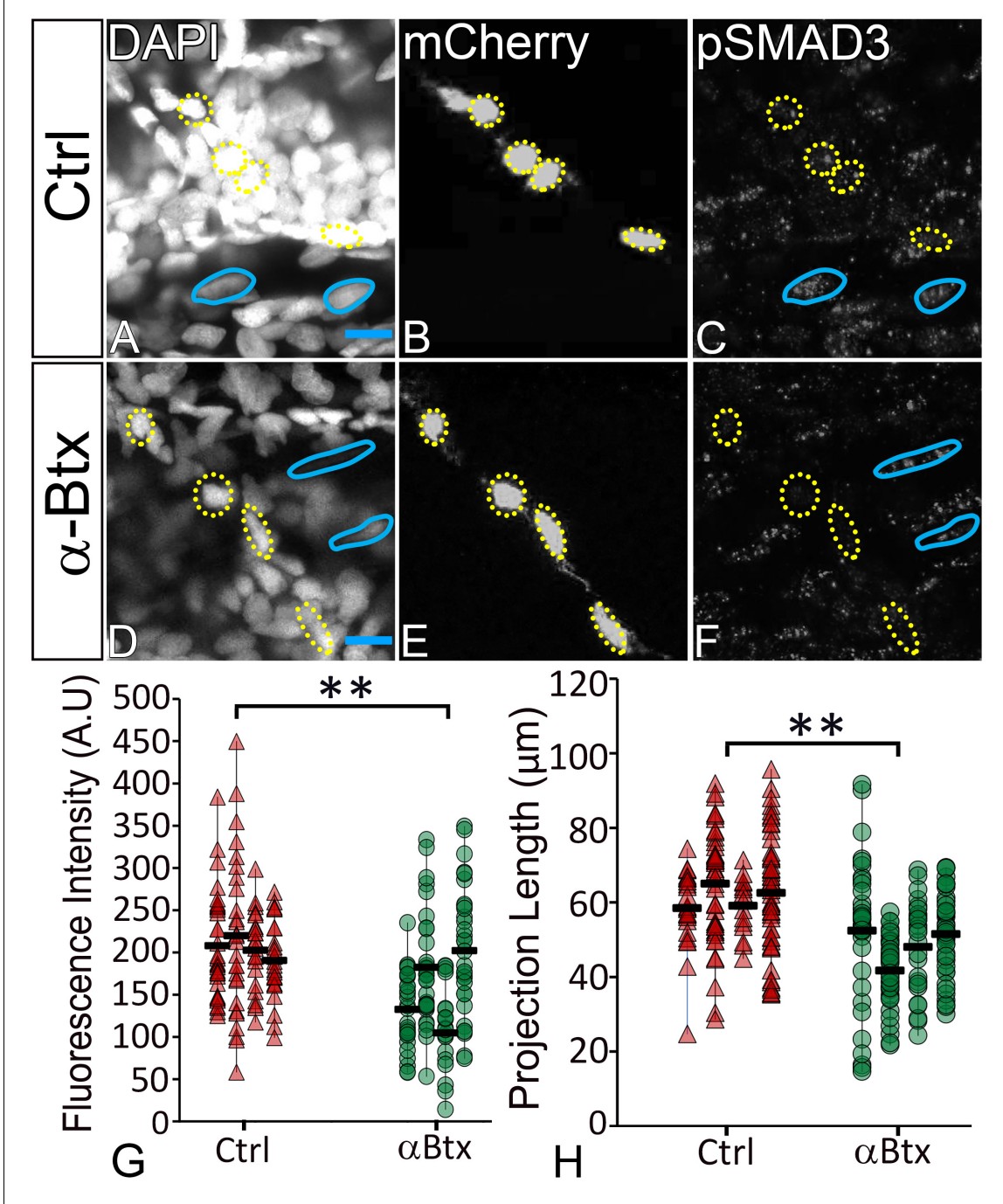

**Figure 6.** TGFβ signaling in tenocytes requires mechanical force. Lateral views of 48 hpf immunostained Tg(*scx:mCherry*) control (**A–C**) and αBtx injected (**D–F**) embryos showing nuclei (DAPI), tenocytes (anti-mCherry) and pSMAD3 (anti-pSMAD3) (marked by yellow-dotted ROI). (**G**) Localization of pSMAD3 was quantified as fluorescent intensity of nuclear pSMAD3 signal and plotted as a dot plot (n = 4, p value was calculated by t-test **<0.005). (**H**) Dot plot shows individual tenocyte projection lengths (p value was calculated by t-test **<0.00005). Representative muscle nuclei are marked by a blue continuous ROI. Scale bar = 10 microns. The measurements used for quantitative analysis and creation of the plots can be accessed from *Figure 6—source data 1* and *Figure 6—source data 2*.

DOI: https://doi.org/10.7554/eLife.38069.031

The following source data and figure supplements are available for figure 6:

**Source data 1.** Measurements of Tenocyte projection length along VMS.

DOI: https://doi.org/10.7554/eLife.38069.036

**Source data 2.** Measurements of tenocyte nuclei pSMAD3 fluorescence intensity along VMS.

*Figure 6 continued on next page*

*Figure 6 continued*

DOI: https://doi.org/10.7554/eLife.38069.037

**Figure supplement 1.** TGFβ signaling is elevated in response to mechanical force.

DOI: https://doi.org/10.7554/eLife.38069.032

**Figure supplement 1—source data 1.** Measurements of tenocyte nuclei pSMAD3 fluorescence intensity along VMS.

DOI: https://doi.org/10.7554/eLife.38069.033

**Figure supplement 2.** Mechanical force regulates expression of genes involved in tendon development.

DOI: https://doi.org/10.7554/eLife.38069.034

**Figure supplement 3.** TGFβ signaling and tenocyte projection integrity affect tendon gene expression.

DOI: https://doi.org/10.7554/eLife.38069.035

## Discussion

Mechanical forces generated by cells adhering to ECM alter their shapes and functions during development, but few studies have investigated the underlying mechanisms in vivo (*Dan et al., 2015*; *Hamada, 2015*; *Ladoux et al., 2015*). Here we show that early developing tenocytes in zebrafish express the tenogenic fate determinant, *scxa*, prior to the differentiation of muscle fibers and respond to the onset of muscle contraction by elongating and extending an array of polarized projections. These projections are disrupted by changes in force as is the corresponding organization of the tendon ECM, which is critical for MTJ function (*Subramanian and Schilling, 2014*). Our results show for the first time in vivo that TGFβ signaling responses induced by mechanical force from muscle contraction correlate with changes in tenocyte morphogenesis and tendon ECM composition during tendon development. These results suggest a novel role for tenocyte projections as force sensors and responders in the feedback between tenocyte and ECM that physically balance responses to mechanical force (*Figure 7*).

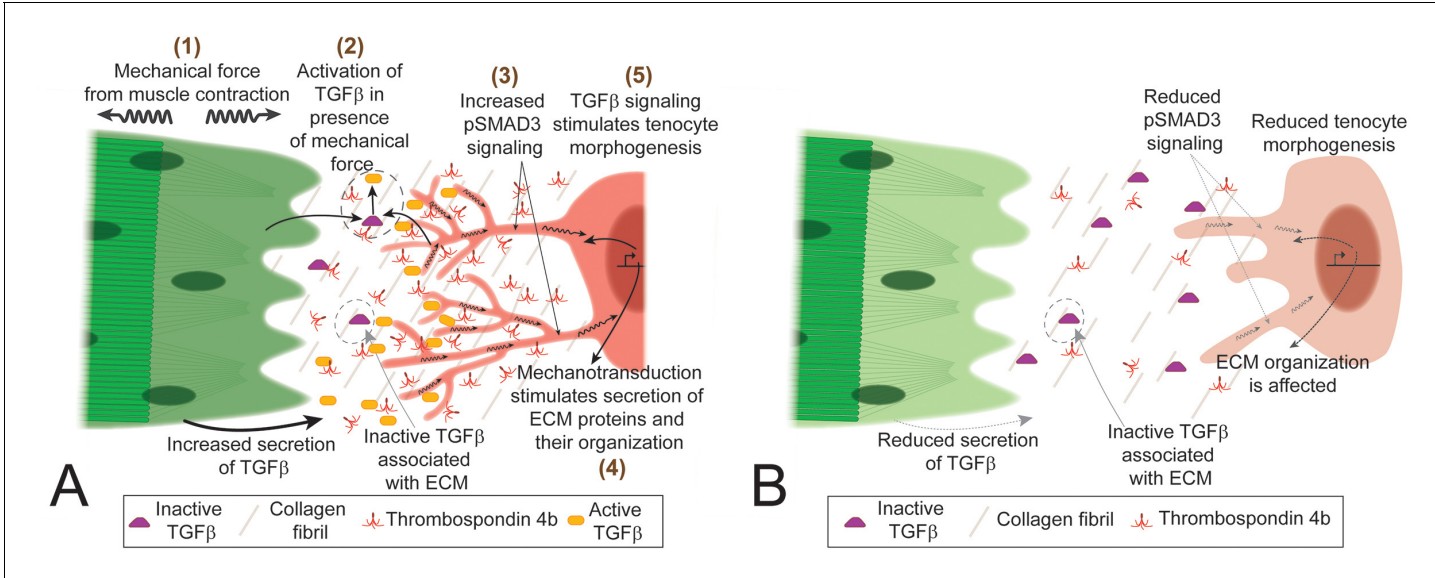

**Figure 7.** TGFβ-mediated mechanotransduction is essential for tenocyte differentiation and morphogenesis. (**A**) In the presence of tensile force from muscle contraction (1) changes in ECM organization and other factors lead to release of active Tgfβ ligand (2). Tgfβ ligand binds to receptors on tenocytes to increase pSMAD3 signaling (3), secretion of ECM components (4) and growth/branching of microtubule rich projections (5). Cartoon depiction of tenocyte morphogenesis in the presence of mechanical force (during onset of muscle contraction in embryonic development or through electrical stimulation of paralyzed embryos). (**B**) In the absence of mechanical force (before onset of muscle contraction or in paralyzed embryos) there is reduced active Tgfβ ligand, pSMAD3 signaling, expression of ECM proteins and growth/branching of projections. Depiction of tenocyte morphogenesis in the absence of mechanical force.

DOI: https://doi.org/10.7554/eLife.38069.038

## Roles for tensional force in tendon morphogenesis

Tendons primarily experience tension from muscle contractions (*Lavagnino et al., 2015*; *Wang, 2006*). In contrast, skeletal cell types (e.g. osteocytes, osteoblasts, chondrocytes) are exposed to compressive forces (*Klein-Nulend et al., 2012*) or shear forces in the case of chondrocytes in joints exposed to fluid flow (*Servin-Vences et al., 2017*).We show that in the absence of tension during development, tenocytes reduce the extent and spread of their projections into the tendon ECM and this can be rescued by a short bout of contraction. Tendon defects and injuries result from dramatic changes in tension experienced either instantly or periodically over extended periods of muscle disuse or overuse (*Franchi et al., 2013*; *Gaut and Duprez, 2016*; *Wang et al., 2012*). Embryonic tenocyte progenitors experience muscle contractions at early stages and must continuously adapt to changes in muscle strength. Our results support the idea that the establishment and adaptation of MTJs occurs in response to mechanical force from muscle contraction and involves both changes in tenocyte morphogenesis and ECM production.

We show that paralysis reduces tenocyte branching and tendon ECM, which can be rescued by restoring muscle contractions through electrical stimulation. Early experiments on developing chick embryos have shown that induced lack of muscle activity (either by lack of neuronal innervation or by injecting paralysis-inducing drugs) negatively affects the growth of associated skeletal structures, suggesting a role for force from muscle contraction as an essential cue for proper growth and differentiation of the skeleton (*Hall and Herring, 1990*; *Hamburger and Waugh, 1940*). During development, the skeleton is exposed to two major types of force – contractile (tension) force from muscles and compressional force (e.g. gravity). A contractile force from muscles has a greater impact on the growth of bones when compared to compression, indicating a primary role for muscle function in guiding the growth of associated skeletal tissues (*Ellman et al., 2014*; *Warden et al., 2013*). Recent studies in paralyzed limbs have shown that the development of a tendon-bone attachment unit, the enthesis, is affected by lack of muscle contraction (*Schwartz et al., 2013*; *Tatara et al., 2014*). Our studies suggest that muscle contraction has a similar role in the development of tendons. Immobilization experiments performed on canine models have shown that mechanical force is required for repair of tendon injuries (*Gelberman et al., 1982*). More recent studies using paralysis and restricted movement have shown that mechanical force has multiple roles in the maintenance of tenogenic gene expression, secretion of tendon ECM, and tenocyte survival (*Gaut et al., 2016*; *Hettrich et al., 2011*; *Maeda et al., 2011*).

## Microtubules are essential for tenocyte projection stability and function

Cellular filopodia and neuronal axons require either F-actin and MTs or both in the formation and maintenance of projections, and new classes of cellular projections are emerging from recent studies such as cytonemes and airinemes (*Bornschlögl, 2013*; *Eom et al., 2015*; *Huang and Kornberg, 2015*; *Witte et al., 2008*). MTs also serve as pathways for trafficking various proteins, RNA, and other intracellular components along projections. Tenocytes in Drosophila rely on a network of polarized MTs for the maintenance of cellular structure and function (*Subramanian et al., 2003*), but similar requirements for cytoskeletal components have not been investigated in vertebrate tendons. Here, we show that zebrafish tenocytes are rich in MTs, which are required to maintain projections. Pharmacological disruption of MTs destabilized the projections without affecting tenocyte cell bodies. This reduced Tsp4b localization suggesting that tenocyte projections both sense force and respond to it by altering ECM organization in an MT-dependent manner. A caveat to this result is that treatment of embryos with Nocodazole causes global destabilization of MT in the entire embryo. Hence, the effects on tenocyte projections and tendon ECM organization could also arise in response to MT destabilization in neighboring muscle fibers, axons and other cells. Similar roles for cellular projections have been observed in pigment cells, where airinemes composed of both F-actin and MTs play a role in long-range signaling by secreting signaling ligands at the tips of their projections (*Eom et al., 2015*). We find that loss of tenocyte projections leads to upregulated expression of *scxa* and *ctgfa* in MT-deficient embryos, suggesting that they revert to a more dedifferentiated state.

## Tenocyte projections are force sensors in the tendon ECM

Previous EM studies of human and rat tendons have described tenocytes projecting into the tendon matrix, but their functional significance has remained unclear (*McNeilly et al., 1996*; *Pingel et al.,*

*2014*). 3D reconstructions from EM studies suggest a role for these projections, referred to as 'fibro-positors' in one study, in secreting collagen fibrils (*Canty et al., 2004*). Analysis of Scx expression in chick and mouse using immunostaining and transgenic reporter lines, respectively, have shown that limb tendons elongate as the musculoskeletal system matures (*Brent et al., 2003*; *Kardon, 1998*; *Pryce et al., 2007*). Our results in zebrafish reveal that such elongated projections are conserved, but quite distinct in different classes of tenocytes. While cranial tenocytes resemble those in the limb in that they extend in parallel to the direction of force, axial tenocytes extend their projections per-pendicular to the plane of muscle contraction (with opposing directions of contractile force) (*Figure 1* and *Figure 1—figure supplement 2*). Based on our results, we propose that these distinct morphol-ogies reflect a more structural, load-bearing role for cranial (and limb) tenocytes, while early larval axial tenocytes in zebrafish function as tension sensors in the myoseptum. Many important questions remain and form the basis of future studies, including why these cells are so polarized and how this mediolateral polarity develops. Consistent with the tension-sensor hypothesis, the timing of the out-growth of tenocyte projections tightly correlates with the onset of muscle contraction. Tension sens-ing projections are observed in other musculoskeletal tissues as osteocytes extend projections into the bone matrix where they are thought to form a network of force sensors (*Cowin et al., 1991*) that modulate bone formation and resorption (*Klein-Nulend et al., 2012*; *Schaffler et al., 2014*). Like-wise, the cues that cause osteoblasts to form these projections as they differentiate into osteocytes remain unknown (*Franz-Odendaal et al., 2006*). Similar to our results with zebrafish tenocytes, mam-malian osteocyte projections increase in density in response to force, consistent with a role as force sensors and responders in both cases.

In both bone and tendon, the ECM undergoes dynamic changes in expression of collagens, fibro-nectin, laminin and MMPs, and this is also the case in the developing somites of zebrafish embryos (*Jenkins et al., 2016*; *Snow and Henry, 2009*). Tenocyte projection formation also correlates with the establishment of tendon ECM. Previous studies have shown remodeling of MTJ ECM between 24 and 48 hpf with a progressive reduction of Fn, which is replaced by increased in levels of Lam at the MTJ (*Jenkins et al., 2016*). Our results suggest that initial production and accumulation of Fn is inde-pendent of force at 24 hpf, when *tsp4b* also shows force-independent expression. The later force-dependent expression and localization of Tsp4b at 48 hpf indicates that dynamic regulation of tendon ECM occurs after the onset of muscle contraction, which suggests a role for mechanical force in the process. Mammalian tenocytes actively sense mechanical force in vitro, resulting in changes in gene expression, cytoskeletal organization and ECM secretion (*Banos et al., 2008*; *Gaut et al., 2016*; *Havis et al., 2016*; *Maeda et al., 2010*; *Maeda et al., 2013*; *Maeda et al., 2011*). This depends, at least in part, on gap junctional complexes that localize to tenocyte projections (*Maeda et al., 2012*). Exercise induces *Tenomodulin* (*Tnmd*) and *Col1a1* expression and tenocyte proliferation in rats (*Eliasson et al., 2009*; *Zhang and Wang, 2013*) and stress induces *COL4A1* and *COL6A1* expression in chick tenocytes (*Marturano et al., 2014*). Despite these changes in gene expression, the molecular mechanisms underlying these cellular signaling responses to force are unclear. Embryonic tenocyte projections in zebrafish end in bouton-like structures close to the dermis (*Figure 1—figure supple-ment 1A,B*), which may act as signaling beacons and ECM secreting centers.

The strong correlation between onset of muscle function, changing myotendinous ECM and teno-cyte morphogenesis suggests a model in which force is transduced through cues from the ECM that induce the formation of projections (*Figure 7A*). Similar processes may underlie the projections of osteocytes and other mesenchymal cell types. Such feedback likely allows tendons to adapt to chang-ing mechanical force during normal development and exercise, as well as in healing and repair of ten-don injuries.

## Mechanical forces and signaling

Our results show for the first time that activation of TGFβ signaling in response to mechanical force is required for tenocyte morphogenesis, in particular the growth and branching of tenocyte projec-tions. Paralyzed embryos (αBTX-injected) lose pSMAD3 expression in tenocytes and projections shorten, which is rescued by restoring muscle contraction. Similarly, pharmacological inhibition of TGFβ receptors reduces pSMAD3 expression and shortens tenocyte projections. Studies of mecha-notransduction have identified several putative signaling pathways involved, depending on the tis-sue, including TGFβ, YAP/TAZ, and Integrins, as well as membrane channels such as TrpV4 and Piezo receptors (*Busch et al., 2017*; *Gumbiner and Kim, 2014*; *Lavagnino et al., 2015*; *Servin-*

*Vences et al., 2017*). Some of these pathways such as TGFβ and YAP/TAZ share intermediate signaling components and targets, which complicates our understanding of their role in mechanotransduction in specific tissues (*Qin et al., 2018*; *Szeto et al., 2016*). This could help explain the modest reduction in expression of *tgfbip* and *ctgfa2* in SB431542-treated embryos, as other mechanotransduction signaling pathways may still function in these embryos to partially maintain expression levels of these genes (*Figure 6—figure supplement 3*) In vitro studies of tenocyte primary cultures and excised tendon tissue have shown elevated TGFβ signaling in response to mechanical load (*Heinemeier et al., 2003*; *Heinemeier et al., 2007*; *Maeda et al., 2013*; *Maeda et al., 2011*; *Yang et al., 2004*) and mice show elevated TGFβ signaling in muscles and tendons following exercise (*Maeda et al., 2011*). These studies suggest that TGFβ signaling, in addition to its earlier role in tenocyte specification (*Havis et al., 2016*), is involved in mechanotransduction in these cells after they differentiate. TGFβ signaling is activated by many factors, including integrins, BMP1 and MMPs which can act on the large latent complex (LLC), to release active TGFβ ligand from the ECM (*Horiguchi et al., 2012*; *Keski-Oja et al., 2004*; *Todorovic et al., 2005*). This could be the critical cue from the ECM that induces and modulates the formation of tenocyte projections. One candidate for initiating these events is Tsp4, since Tsps can activate TGFβ signaling by destabilizing latency-associated peptide (LAP) (*Bailey Dubose et al., 2012*). The tendon matrix is rich in MMPs and Tsps, which dynamically change in composition and activity depending on mechanical force (*Jenkins et al., 2016*; Popov et al., 2015; *Subramanian and Schilling, 2014*). The dynamic reductions in Tsp4b that we have shown in response to paralysis could fail to activate latent TGFβ in the tendon ECM. Furthermore, because we see reductions in Tsp4b expression in paralyzed embryos, our results support a model where force triggers TGFβ signaling leading to increased expression of Tsp4b, which in turn activates TGFβ expression, creating a positive feedback loop (*Figure 7A,B*). Transection of tendons or injection of botulinum toxin (Botox) to induce paralysis in mice causes tenocyte death and reduced expression of tenogenic genes (*Maeda et al., 2011*). In contrast, we observe neither cell death nor significant changes in tenogenic gene expression in paralyzed (*cacnb1* mutant) zebrafish embryos until 5 dpf, several days after tenocyte differentiation. We interpret such a response as a separate response to prolonged disuse rather than an adaptation to force.

These studies have shown a direct relationship between mechanical force and tendon development through TGFβ signaling in tenocytes. How do tenocyte progenitors begin the process of elongation and growth of projections? What are the roles of these projections during tendon embryonic development and in adult tendons? Osteocytes are known to induce repair pathways in bone when cracks or stress damage their processes (*Dooley et al., 2014*; *Mulcahy et al., 2011*). Do tenocyte projections perform a similar role in tendon repair? These are some of the questions that need to be addressed in the field of tendon biology. Understanding the relationship between force and tendon development is essential for developing effective treatment strategies that include engineering tendons to treat tendon injuries. Force sensing projections that allow cells to adjust their surrounding ECM, such as those we have described in tenocytes, may also be a more general feature of cells, particularly within the musculoskeletal system.

# Materials and methods

**Key resources table**

| Reagent type (species) or resource | Designation | Source or reference | Identifiers | Additional information |
|---|---|---|---|---|
| Antibody | Rabbit anti Tsp4b | Schilling lab | RRID: AB_2725793 | 1:500 dilution |
| Antibody | Mouse anti Myosin heavy chain | DSHB | Cat# A4.1025, RRID: AB_528356 | 1:250 dilution |
| Antibody | Chicken anti GFP | Abcam | Cat# ab13970, RRID: AB_300798 | 1:1000 dulution |
| Antibody | Rat anti mCherry | Molecular Probes | Cat# M11217, RRID: AB_2536611 | 1:500 dilution |
| Antibody | Rabbit anti Fibronectin | Abcam | Cat# ab2413, RRID: AB_2262874 | 1:200 dilution |

*Continued on next page*

*Continued*

| Reagent type (species) or resource | Designation | Source or reference | Identifiers | Additional information |
|---|---|---|---|---|
| Antibody | Rabbit anti Laminin | Abcam | Cat# ab11575, RRID: AB_298179 | 1:200 dilution |
| Antibody | Rabbit anti pSMAD3 | Antibodies-online | Cat# ABIN1043888, RRID: AB_2725792 | 1:500 dilution |
| Antibody | Alexa Fluor 488 conjugated Donkey anti Chicken IgY | Jackson Immunoresearch | Cat# 712-586-153 | 1:1000 dulution |
| Antibody | DyLight 549 conjugated Donkey anti Rabbit IgG | Jackson Immunoresearch | Cat# 711-506-152, RRID: AB_2616595 | 1:1000 dulution |
| Antibody | Alexa Fluor 488 conjugated Donkey anti Rabbit IgG | Jackson Immunoresearch | Cat# 711-545-152, RRID: AB_2313584 | 1:1000 dulution |
| Antibody | Cy5 conjugated anti Mouse IgG | Jackson Immunoresearch | Cat# 115-176-071 | 1:1000 dulution |
| Antibody | Alexa Fluor 594 conjugated Donkey anti Rat IgG | Jackson Immunoresearch | Cat# 712-586-153, RRID: AB_2340691 | 1:1000 dulution |
| Antibody | Alexa Fluor 488 conjugated anti Mouse IgG | Jackson Immunoresearch | Cat# 715-546-150; RRID: AB_2340849 | 1:1000 dulution |
| Antibody | DiAmino PhyenylIndole (DAPI) | Invitrogen | Cat# D1306, RRID: AB_2629482 | 1:1000 dulution |
| Cell line (*E. coli*) | Chemically competent DH5alpha cells | Schilling Lab | | |
| Chemical compound, drug | SB431542 | Tocris | Cat# 1614, SID: 241182574 | 50 mM stock solution, 10 μM final concentration |
| Chemical compound, drug | Nocodazole | Sigma-Aldrich | Cat#1404, SID: 24278535 | 33 mM stock solution, 0.33 mM final concentration |
| Chemical compound, drug | Trizol | Invitrogen | Cat# 15596018 | |
| Chemical compound, drug | 3-aminobenzoic acid ethyl ester methanesulfonate | Sigma-Aldrich | Cat# A5040, SID: 329770864 | |
| Commercial assay or kit | mMessage mMachine T7 ultra transcription kit | Ambion | Cat# AM1345, RRID: SCR_016222 | |
| Commercial assay or kit | mMessage mMachine T3 transcription kit | Ambion | Cat# AM1348, RRID: SCR_016223 | |
| Commercial assay or kit | mMessage mMachine SP6 transcription kit | Ambion | Cat# AM1340, RRID: SCR_016224 | |
| Commercial assay or kit | Protoscript II first strand cDNA synthesis kit | New England Biolabs | Cat# E6560, RRID: SCR_016225 | |
| Commercial assay or kit | Luna universal qPCR master mix | New England Biolabs | Cat# M3003, RRID: SCR_016226 | |
| Commercial assay or kit | Direct-zol RNA Miniprep | Zymo Research | Cat# R2061, RRID: SCR_016227 | |
| Commercial assay or kit | QX200 EvaGreen 653 ddPCR Supermix | Bio-Rad | Cat# 1864033 RRID: SCR_008426 | |
| Commercial assay or kit | QX200 Droplet Generation Oil for EvaGreen | Bio-Rad | Cat# 1864005, RRID: SCR_008426 | |
| Commercial assay or kit | QX200 Droplet Generator | Bio-Rad | Cat# 1864002, RRID: SCR_008426 | |
| Commercial assay or kit | QX200 Droplet 657 Reader | Bio-Rad | Cat# 1864003, RRID: SCR_008426 | |

*Continued on next page*

*Continued*

| Reagent type (species) or resource | Designation | Source or reference | Identifiers | Additional information |
|---|---|---|---|---|
| Commercial assay or kit | Qubit SSDNA assay kit | Invitrogen | Cat# Q10212, SCR_008817 | |
| Commercial assay or kit | Qubit 2.0 fluorometer | Invitrogen | Cat# Q32866, SCR_008817 | |
| Gene (*Danio rerio*) | Tg(*scx:mCherry*) | Galloway lab | N/A | |
| Gene (*Danio rerio*) | *Cacnb1*$^{+/-}$ | Schilling lab | N/A | |
| Sequence-based reagent | Primers for RT-PCR, see Table S1 | This paper | N/A | 0.5 μM final concentration |
| Recombinant DNA reagent | *pmtb-t7-alpha-bungarotoxin* | Addgene | Cat# 69542, RRID: SCR_002037 | |
| Recombinant DNA reagent | *pIRESneo-EGFP-alpha tubulin* | Addgene | Cat# 12298, RRID: SCR_002037 | |
| Recombinant DNA reagent | *pmEGFP-Lifeact-7* | Addgene | Cat# 54610, RRID: SCR_002037 | |
| Software, algorithm | Simple Neurite Tracer | Fiji | | |

## Zebrafish transgenics and mutants

Tg(*scx:mCherry*) transgenics were generated by injecting a BAC construct (CH211-251g8) containing mCherry ORF inserted in frame after the start codon of the *scxa* gene (*McGurk et al., 2017*). A new mutant allele of *cacnb1* was identified in a forward genetic screen and outcrossed with Tg(*scxa:mCherry*) to create a *cacnb1*;Tg(*scxa:mCherry*) line. All embryos were raised in embryo medium at 28.5°C (*Westerfield, 2007*), and staged as described previously (*Kimmel et al., 1995*). Craniofacial muscles and cartilages were labeled as described previously (*Schilling and Kimmel, 1997*). Adult fish and embryos were collected and processed in accordance with approved UCI-IACUC guidelines.

## mRNA injections and drug treatments

A *Pmtb-t7-alpha-bungarotoxin (αBtx)* vector (Megason lab, Addgene, 69542) was used to synthesize αBtx mRNA following a previously published protocol and injected into Tg(*scx:mCherry*) embryos at the 1–2 cell stage (*Subramanian and Schilling, 2014*; *Swinburne et al., 2015*). A *pIRESneo-EGFP-alpha Tubulin* plasmid (Wadsworth lab, Addgene, 12298) was used to synthesize EGFP-α Tubulin mRNA following a previously published protocol and injected into Tg(*scx:mCherry*) embryos at the 1–2 cell stage (*Rusan et al., 2001*; *Subramanian and Schilling, 2014*).

A stock solution of 50 mM SB431542 (Tocris 1614, SID: 241182574)), a selective inhibitor of TGFβ type I receptor was prepared in DMSO (Fisher Scientific D1281, SID: 349996472) and diluted to a final working concentration of 10 μM in embryo medium. Embryos were incubated in 10 μM SB431542 for 12 hr. Treated embryos were rinsed in pre-warmed (28.5°C) embryo medium before fixation for immunostaining or RNA extraction.

A stock solution of 33 mM Nocodazole (Sigma M1404, SID: 336851328), an inhibitor of tubulin polymerization, was prepared in DMSO and diluted to a final working concentration of 0.33 mM in embryo medium. Embryos were incubated in 0.33 mM Nocodazole for 12 hr at 28.5°C. Treated embryos were either mounted for live imaging or fixed for immunostaining.

## RT-PCR

Whole embryo RNA was extracted from control and paralyzed embryos collected at 48 hpf according to standard protocols using Trizol (Invitrogen 15596018) and Direct-zol RNA MinipPrep kits (Zymo Research R2061). RNA concentration was normalized between samples and used as a template for cDNA synthesis. cDNA was synthesized with oligodT primers using the standard protocol of ProtoScript II First Strand cDNA Synthesis Kit (NEB E6560). The synthesized cDNA was diluted to 1:20 and used as a template for RT-PCR using the protocol for the Luna Universal qPCR master mix (NEB M3003S). The primers used for RT-PCR are listed in *Table 1*. The reaction was run on a Light-Cycler 480 II Real time-PCR Instrument (Roche) and analyzed using LightCycler 480 Software. Each

**Table 1.** List of primer sequences used for RT-PCR.

| Name | Sequence | Gene |
|---|---|---|
| rpl13a-fp-qpcr | TCTGGAGGACTGTAAGAGGTATGC | *ribosomal protein L13a* |
| rpl13a-rp-qpcr | AGACGCACAATCTTGAGAGCAG | |
| rps13-fp-qpcr | ATAGGCGAAGTGTCCCCACA | *ribosomal protein S13* |
| rps13-fp-qpcr | CAGTGACGAAACGCACCTGA | |
| scxa-fp-qpcr | GGAGAACTCGCAGCCCAAA | *scleraxis A* |
| scxa-rp-qpcr | AATCCCTTCACGTCGTGGTTT | |
| tsp4b-fp-qpcr | ACAATCCACGAGACAACAGC | *thrombospondin 4b* |
| tsp4b-rp-qpcr | GCACTCATCCTGCCATCTCT | |
| ctgfa-fp-qpcr | CTTTACTGTGACTACGGCTCC | *connective tissue growth factor a* |
| ctgfa-rp-qpcr | ACAACTGCTCTGGAAAGACTC | |
| tgfbip-fp-qpcr | CCCCAATGTTTGTGCTATGC | *tgfβ induced peptide* |
| tgfbip-rp-qpcr | CTCCAATCACCTTCTCATATCCAG | |

DOI: https://doi.org/10.7554/eLife.38069.039

qPCR experiment was designed with triplicates of reactions for every biological sample and two biological samples were used for each analysis (*Subramanian and Schilling, 2014*).

## ddPCR

cDNA was prepared from whole embryo RNA using the standard protocol of ProtoScript II First Strand cDNA Synthesis Kit (NEB E6560). The cDNA concentration was determined following standard protocol and reagents from the Qubit SSDNA assay kit (Invitrogen Q10212) and fluorescence was read on a Qubit 2.0 fluorometer (Invitrogen Q32866). A total concentration of 1 ng was used from each sample to prepare 20 ml of ddPCR reaction following the instructions and reagents from QX200 EvaGreen ddPCR Supermix (Bio-Rad 186–4033).Primers for the PCR are listed in *Table 1*. The droplets were generated using QX200 Droplet Generation Oil for EvaGreen (Bio-Rad 1864005) on a QX200 Droplet Generator (Bio-Rad 1864002). The PCR reaction was run on a standard thermocycler under standard cycling conditions. Following the PCR the droplets were analyzed using QX200 Droplet Reader (Bio-Rad 1864003). The data were analyzed using QuantaSoft Analysis Pro Software.

## Muscle stimulation

Electrical stimulation was used to induce muscle contraction, as previously described (*Subramanian and Schilling, 2014*). Both αBtx injected and control embryos or larvae were anaesthetized with Tricaine (ethyl 3-aminobenzoate methanesulfonate, Sigma A5040, SID: 329770864), placed on a silicone plate in embryo medium and stimulated for 2 min at 20V, 6 msec duration, 4 pulses/sec frequency and 6 msec delay between successive pulses. With these settings neither control nor paralyzed embryos showed any muscle detachment. Embryos were allowed to recover in embryo medium for 12 hr and further processed for immunostaining or RT-PCR.

## Whole embryo immunohistochemistry

All embryos used for immunofluorescence experiments were fixed in 4% neutral pH buffered paraformaldehyde (PFA) for 2 hr at room temperature (25℃) or overnight at 4℃. The embryos were washed with 1X Phosphate Buffered Saline (PBS, CID: 24978514) and permeabilized with cold acetone (Fisher Scientific A94, SID: 349996362) for 15 min at −20℃. Following permeabilization, they were rehydrated in PBDT (PBS with 2% DMSO and 1% Triton X-100 (Sigma T9284)) and processed according to a standard antibody staining protocol. Primary antibodies used: rabbit anti-Tsp4b (1:500)(RRID: AB_2725793), mouse anti-myosin heavy chain (MHC) (Developmental Hybridoma - 1:250, A1025, RRID: AB_528356), chicken anti-GFP (Abcam – 1:1000, ab13970, RRID: AB_300798), rat monoclonal anti-mCherry (Molecular Probes −1:500, M11217, RRID: AB_2536611), rabbit anti-Laminin (Abcam – 1:200, ab11575, RRID: AB_298179), rabbit anti-Fibronectin (Abcam – 1:200,

 

ab2413, RRID: AB_2262874 and rabbit anti-pSMAD3 (Antibodies-online – 1:500, ABIN1043888, RRID: AB_2725792). DiAmino PhenylIndole (DAPI) (Invitrogen – 1:1000, D1306, RRID: AB_2629482) was used to mark cell nuclei. Preabsorbed secondary antibodies were all obtained from Jackson ImmunoResearch and used for indirect immunofluorescence at 1:1000, including: Alexa Fluor 488 conjugated donkey anti-mouse IgG (715-546-150, RRID: AB_2340849), DyLight 549 conjugated donkey anti-rabbit IgG (711-506-152, RRID: AB_2616595), Alexa Fluor 488 conjugated donkey anti-rabbit IgG (711-545-152, RRID: AB_231358), Cy5 conjugated Goat anti-mouse IgG (115-176-071), Alexa Fluor 594 conjugated donkey anti-rat IgG (712-586-153, RRID: AB_2340691), and Alexa Fluor 488 conjugated donkey anti-chicken IgY (703-486-155). After staining, embryos were mounted in 1% low melt agarose in PBS and imaged.

## Microscopy and image analysis

Embryos processed for fluorescent immunohistochemistry were imaged using a Nikon A1 confocal system with an Nikon Eclipse T*i* inverted microscope using a CFI Plan Apochromat VC 60XC (water immersion) objective. Confocal stacks were analyzed using Image J software. The depth-coded 3D reconstructions were created using Nikon software (NIS-Elements AR 4.60.00 64-bit). To better visualize tenocyte projections along the Z-axis, the 3D reconstructed image was rotated to about 45°. The length of projections was measured using the Neurite Tracer plugin on Image J.

## Statistical analysis

Sample size and number of data points required for each experiment were determined using a power analysis calculator (www.powerandsamplesize.com). The embryos were collected from a single tank of fish and processed for injection and downstream stimulation together to minimize variation introduced during handling. Fixation and staining of embryos were also performed together for all samples in a given experiment. Imaging of embryos within each experiment was performed with identical parameters. In order to control for variation in position of tenocyte cell bodies and antibody penetrance variation, projection length, fluorescence intensity of ECM proteins and pSMAD3 were always measured in the ventral half of the VMS in somites 16–19. In experiments where a normal distribution was not present, we analysed the significance using a Wilcoxon Rank Sum test. In datasets involving two samples of unequal variance, a t-test was used. In experiments with more than two experimental conditions, an ANOVA single-factor analysis was performed with posthoc multiple comparisons using Tukey method on R. Data were also quantified and analyzed separately by two of the authors to account for user bias and they obtained similar results. Fluorescence Intensity (FI) to quantify protein localization was measured as described previously (*Subramanian and Schilling, 2014*).

## Acknowledgements

We thank members of the Schilling lab for comments on the manuscript and I Gehring for fish care. We also thank Z Wunderlich, L Li and R Bautista for statistics advice. This work was supported by NIH grants R01 AR67797, R21 AR62792 and R01 DE013828, to TFS.

## Additional information

### Funding

| Funder | Grant reference number | Author |
| --- | --- | --- |
| National Institutes of Health | R01 AR67797 | Thomas Friedrich Schilling |
| National Institutes of Health | R01 DE013828 | Thomas Friedrich Schilling |
| National Institutes of Health | R21 AR62792 | Thomas Friedrich Schilling |
| National Institutes of Health | R00 HD069533 | Jenna Lauren Galloway |
| National Institutes of Health | R01 AR074541 | Jenna Lauren Galloway |

The funders had no role in study design, data collection and interpretation, or the decision to submit the work for publication.

## Author contributions
Arul Subramanian, Conceptualization, Data curation, Formal analysis, Supervision, Validation, Investigation, Visualization, Methodology, Writing—original draft, Writing—review and editing; Lauren Fallon Kanzaki, Investigation, Visualization, Methodology, Writing—review and editing; Jenna Lauren Galloway, Resources, Writing—review and editing; Thomas Friedrich Schilling, Conceptualization, Resources, Formal analysis, Supervision, Funding acquisition, Investigation, Writing—original draft, Project administration, Writing—review and editing

## Author ORCIDs
Arul Subramanian http://orcid.org/0000-0001-8455-6804
Lauren Fallon Kanzaki https://orcid.org/0000-0002-9564-2385
Jenna Lauren Galloway http://orcid.org/0000-0003-3792-3290
Thomas Friedrich Schilling http://orcid.org/0000-0003-1798-8695

## Ethics
Animal experimentation: This study was performed in strict accordance with the recommendations in the Guide for the Care and Use of Laboratory Animals of the National Institutes of Health. All of the animals were handled according to approved institutional animal care and use committee (IACUC) protocols (#2000-2149) of the University of California, Irvine. Embryos were anesthetized with tricaine before stimulation assays.

## Decision letter and Author response
Decision letter https://doi.org/10.7554/eLife.38069.042
Author response https://doi.org/10.7554/eLife.38069.043

# Additional files
## Supplementary files
• Transparent reporting form
DOI: https://doi.org/10.7554/eLife.38069.040

## Data availability
All data generated or analyzed during this study are included in the manuscript and supporting files.

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
