## [Decision Letter]

Thank you for submitting your manuscript entitled "Mechanical force regulates tendon extracellular matrix organization and tenocyte morphogenesis through TGFβ signaling" for peer review at *eLife*. Your article has been evaluated by three peer reviewers, and the evaluation has been overseen by a Reviewing Editor and by Anna Akhmanova as the Senior Editor. The reviewers have discussed the reviews with one another, and their discussion has raised a number of issues (elaborated below) that would need to be addressed through revisions to your manuscript.

Summary:

In this manuscript, Subramanian and colleagues investigate the role of tenocyte projections in the development of the muscle-tendon junction. Their newly generated *scx:mCherry* line has unlocked the ability to image the tenocyte and its interactions in living muscle tissue, providing unique insights in to the in vivo cell biology of tenocyte maturation. Using this line, they first observe the development of tenocytes and note that these cells form very long cellular protrusions that are microtubule-based. Through a clever set of experiments, they show that the protrusions require muscle contraction for their full extension. Impairing the full development of these tenocyte projections reduces the amount of the myotendinous protein Tsp4b and alters its localization pattern. This led the authors to ask if these projections sense mechanical force through TGFβ, a known latent ECM cue activated by mechanical stress. Similar to blocking muscle contraction, inhibition of TGFβ signaling also reduces pSMAD3 levels and reduces the length of tenocyte projections. Consistent with the idea that muscle contraction activates TGFβ signaling in tenocytes, the authors find that tenocyte genes are also down regulated when muscle contraction is blocked or TGFβ signaling is blocked. Altogether, this is an interesting study with spectacular imaging that has the potential to be an important paper in the field of tendon extracellular matrix biology. However, there are a few concerns that should be addressed before publication.

Essential revisions:

1) The Introduction does not quite accurately describe the state of the field. For example, the authors state: "Despite recent insights into the nature of such responses, few studies have investigated how cells adapt to force and alter the ECM landscape to strengthen or weaken it accordingly". Collagen synthesis at tendons has been linked to mechanical loading and inactivity decreasing collagen turnover has been identified since the early 2000s. (One example is "Role of Extracellular Matrix in Adaptation of Tendon and Skeletal Muscle to Mechanical Loading", 2004). It would be helpful to modify the text to highlight that there is a fair amount known about how cells in culture respond to force and change the matrix accordingly, but less is known in vivo (with appropriate references).

2) The manuscript seems to be written for a reader who is quite familiar with the zebrafish system. The impact of the manuscript would be much higher if the authors made it more accessible for a broader range of readers (explaining the difference between what they are calling embryonic and larval development, what are the horizontal and vertical myosepta, what is a hemisegment, etc.). Also, while there is controversy amongst zebrafish investigators at which point the somite boundaries should be called vertical myosepta versus myotendinous junctions, it is also potentially true that the impact outside the field would be higher if the authors decided to call them MTJs.

3) The authors see a reduction in the length of the projections from 70 μm to 50 μm. They imply that this is causal for mechanical sensing and gene expression by tenocytes. However, there is no evidence for this. They should tone down this statement a little. They should also clarify why they think that a reduction by 20% or so is consequential to the MTJ.

4) Nocodazole treatment is a very harsh way of assessing the role of microtubules in tenocyte projection formation and mechanical tension. Treatment with Noco for 12 hours will most likely block many biological processes besides tenocyte projection formation, and some of these processes could indirectly impact tenocytes. Was this blocking performed for 12 hours directly before fixing? Why was such a long time chosen? Do shorter treatments affect projections? Is there a strategy that the authors could use to block microtubule function in tenocytes specifically? If not, they should certainly state the caveats of whole-embryo treatment with nocodazole, and they should modify their interpretations accordingly, especially regarding their claim that microtubules are required for tenocyte projection formation and tenocyte gene expression.

5) LA treatment should be done on its own. Maybe the tenocyte projections are dependent on both MT and F-actin? The no-enhancement observation does not exclude this. If true, this would also give the authors genetic tools to specifically affect actin and thus projections in the tenocytes.

6) Ubiquitous Tsp4b-GFP expression and force: The authors postulate that Tsp4b is secreted by tenocytes upon force generation by muscles. However, this experiment is testing only if force affects the localization of Tsp4-GFP/Tsp4b but not its secretion by tenocytes. In fact, the presence (albeit diffuse) of Tsp4b protein in αBTX embryos shows that force does not induce secretion of Tsp4b per se (though it might enhance it).

7) Depth-coding color scale is inconsistent, sometimes pink=0μm, sometimes blue=0μm. It is important to fix this because without a consistent scale one cannot judge changes in tenocyte projection length. Also, please adjust scale such that the soma of the tenocytes is always at the same depth/color to judge length defects.

8) Quantification of antibody fluorescence is tricky/error-prone. Ratio images between mCherry signal from *scx:mCherry* and pSMAD3 signal would be helpful to visualize changes. Ratio image of panels C divided by B and panels G divided by F would be informative in Figure 5 and 6.

9) Force-dependent gene expression: αBTX looks convincing but SB431542-treated embryos show a mild reduction only. Nocodazole affects gene expression inconsistent with hypothesis/conclusion (*scx* goes up, *ctgfa2* goes up – maybe because nocodazole just makes the embryos very sick?). If force induces TGFβ signaling which induces gene expression, then the changes in gene expression should be roughly equivalent in *αBtx* (no force) and SB431542-treatments (no TGFβ signaling). This is not the case, so force does not only act through TGFβ signaling (or the inhibitor does not fully inhibit – unlikely though since pSMAD3 is more reduced than in *αBtx* injections). Since qPCR is a global (and fairly noisy) assay, is there another way to look at gene expression (antibodies, reporter transgenes)? If so, *scx:mCherry* reporter/antibody embryos could be analyzed in the different conditions and mCh could serve as a reference/normalization (not in the case of nocodazole though).

10) If αBTX injection nearly completely suppresses *tsp4b*mRNA expression by qPCR, why is there Tsp4b protein (Figure 3—figure supplement 1) in these embryos? The reduction in Tsp4b protein is by far not as striking. Could this be clarified? Also, is there a way to control for both initial levels of expression and the rate of decline of mRNA in different embryos? Is it possible that electrical stimulation not only results in restoring force but also signaling?

11) The authors contend that ECM organization of the vertical myosepta is under the control of tenocytes. However, they only look at one specialized component of the ECM, *thrombospondin4*. It would be helpful if Figure 3 could better address the issue of tenocytes' contribution to ECM organization as a whole. Although the authors have previously shown that Thrombospondin4 is an important regulator of muscle cell adhesion, it is unlikely to be the major component of the muscle extracellular matrix. The majority of the matrix is already secreted directly from the muscles themselves, and initial muscle attachment occurs in the absence of tenocytes. How do tenocytes reinforce or mature the ECM generally? Would it be feasible to stain for two or more major ECM components in the vertical myosepta, such as laminin, fibronectin or collagen? TEM could also be an excellent readout.

12) Image analysis/stats: The authors are lauded for attempting to quantify their data. However, the details provided do not necessarily provide confidence in how the data were analyzed.

a) The image analysis section does not specifically address the issue of blinding, and certainly many of the measurements done depend tremendously on user input.

b) How exactly were projections segmented and branch length was quantified? Was this done manually? If images were segmented in Fiji then the parameters of masking, etc. should be detailed.

c) The t-test is appropriate for normal distributions, was this the case with the data obtained?

d) The authors said power analyses were done but still 3 embryos per conditions seems quite low. Also, how were the 50 data points/embryo chosen? Were they always from the same anterior-posterior location? There could be a great deal of subjectivity in the choosing of those data points as well and theoretically they should have been randomly chosen.

e) Given that the relevant details were not provided, it is likely that the analysis and stats were done in a less than ideal fashion. Rather than spend 2 months redoing them, one option would be for the authors to provide supplemental data with multiple image examples of each experiment.

---

## [Author Response]

Essential revisions:1) The Introduction does not quite accurately describe the state of the field. For example, the authors state: "Despite recent insights into the nature of such responses, few studies have investigated how cells adapt to force and alter the ECM landscape to strengthen or weaken it accordingly". Collagen synthesis at tendons has been linked to mechanical loading and inactivity decreasing collagen turnover has been identified since the early 2000s. (One example is "Role of Extracellular Matrix in Adaptation of Tendon and Skeletal Muscle to Mechanical Loading", 2004). It would be helpful to modify the text to highlight that there is a fair amount known about how cells in culture respond to force and change the matrix accordingly, but less is known in vivo (with appropriate references).

1a) We have modified the text to clarify these points and include additional references. However, while in vitro studies in collagen gels have shown that force is necessary for tendon ECM turnover, exercise paradigms have given mixed results on the effect of force on tendon ECM. In general, a systematic analysis of the role of force on developing tendons in vivo is lacking, particularly its influence on tenocyte differentiation.

1b) Introduction, first paragraph: We have added references that appropriately show the role of force in ECM organization (Maeda et al., 2011; Ng et al., 2014).

1c) Introduction, first paragraph: We have also added the word “in vivo” to stress the fact that there are very few in vivo studies that have investigated the role of force in ECM organization and cellular morphogenesis.

2) The manuscript seems to be written for a reader who is quite familiar with the zebrafish system. The impact of the manuscript would be much higher if the authors made it more accessible for a broader range of readers (explaining the difference between what they are calling embryonic and larval development, what are the horizontal and vertical myosepta, what is a hemisegment, etc.). Also, while there is controversy amongst zebrafish investigators at which point the somite boundaries should be called vertical myosepta versus myotendinous junctions, it is also potentially true that the impact outside the field would be higher if the authors decided to call them MTJs.

2a) We have explained zebrafish-specific developmental time points and anatomical terms.

2b) Subsection “Tenocytes elongate with the onset of muscle contraction”, first paragraph: Explains the difference between embryonic and larval time points.

2c) Subsection “Tenocytes elongate with the onset of muscle contraction”, first paragraph: Explains the terms VMS and HMS.

2d) Subsection “Tenocytes elongate with the onset of muscle contraction”, first paragraph: We have removed the term hemisegment to avoid confusion and have used the term MTJ to refer to muscle attachment sites.

3) The authors see a reduction in the length of the projections from 70 μm to 50 μm. They imply that this is causal for mechanical sensing and gene expression by tenocytes. However, there is no evidence for this. They should tone down this statement a little. They should also clarify why they think that a reduction by 20% or so is consequential to the MTJ.

3a) We find a strong correlation between force-induced changes in both projection length and branching with associated gene expression, including ECM proteins we know are vital for MTJ integrity. However, proving experimentally that this is causal is extremely difficult, and we have toned down our arguments accordingly. Reduced length implies that projections sense force changes and respond by penetrating and traversing through the ECM. While the roles of such projections in tenocytes are not well understood, similar responses to force have been shown for osteocytes, as we already mention in the text. We include data on projection length, number, and branching at additional stages. While 20% may not seem like much, this combined with reduced branching leads to many fewer projections per unit area, which we have quantified and added to the Results.

3b) Subsection “Tenocyte elongation requires muscle contraction”: We have performed further analyses to show that in addition to length of projections and branching complexity, the density of projections along the VMS is also affected in a significant manner. We include the data as Figure 2—figure supplement 1.

3c) Subsection “Tenocyte elongation requires muscle contraction”: We also stress that our results only show a strong correlation between mechanical force and tenocyte morphogenesis.

4) Nocodazole treatment is a very harsh way of assessing the role of microtubules in tenocyte projection formation and mechanical tension. Treatment with Noco for 12 hours will most likely block many biological processes besides tenocyte projection formation, and some of these processes could indirectly impact tenocytes. Was this blocking performed for 12 hours directly before fixing? Why was such a long time chosen? Do shorter treatments affect projections? Is there a strategy that the authors could use to block microtubule function in tenocytes specifically? If not, they should certainly state the caveats of whole-embryo treatment with nocodazole, and they should modify their interpretations accordingly, especially regarding their claim that microtubules are required for tenocyte projection formation and tenocyte gene expression.

4a) We have done shorter nocodazole treatments [our original experiments were based on published studies that treated up to 6 hrs (Mendieta-Serrano et al., 2013)] but these did not show strong effects on tenocytes and treated embryos appeared similar to siblings.

Subsection “Microtubules are essential for tenocyte projection stability and function”: We have added a caveat in the Discussion acknowledging the global effects of Nocodazole on the whole embryo and the possibility of indirect effects on tendon ECM and tenocyte morphology.

5) LA treatment should be done on its own. Maybe the tenocyte projections are dependent on both MT and F-actin? The no-enhancement observation does not exclude this. If true, this would also give the authors genetic tools to specifically affect actin and thus projections in the tenocytes.

5a) Injection of LifeAct DNA results in high mortality. However, in the few embryos that survive F-actin is well labeled elsewhere but there is no visible fluorescence in tenocyte projections, suggesting that they lack actin fibrils. We have performed LifeAct treatments alone and did not observe any effect on tenocytes at low doses. In addition, several attempts in injecting F-actin-GFP showed no expression in tenocytes. Hence, we are confident that F-actin is not a major player in tenocyte projection growth and maintenance. To avoid further confusion, we have removed the Latrunculin results from the manuscript and the figure.

6) Ubiquitous Tsp4b-GFP expression and force: The authors postulate that Tsp4b is secreted by tenocytes upon force generation by muscles. However, this experiment is testing only if force affects the localization of Tsp4-GFP/Tsp4b but not its secretion by tenocytes. In fact, the presence (albeit diffuse) of Tsp4b protein in αBTX embryos shows that force does not induce secretion of Tsp4b per se (though it might enhance it).

6a) Even though some Tsp4b protein remains at MTJs in αBTX-treated paralyzed embryos, this is likely produced by myoblasts prior to the onset of muscle contraction. We have previously shown that Tsp4b is first expressed throughout the myotome at the 18-somite stage (20 hpf), prior to muscle differentiation, and only later becomes restricted to tenocytes after 24 hpf (Subramanian and Schilling, 2014). Presumably any protein synthesized at these early stages later localizes to MTJs. Consistent with this, transplants of muscle progenitors locally produce Tsp4b, which can rescue muscle attachments (Subramanian and Schilling 2014). Later it appears that tenocytes take over production of Tsp4b and other ECM proteins to strengthen and maintain attachments.

6b) To further clarify this force-dependent versus -independent Tsp4b expression we have performed qPCR on αBTX-treated paralyzed embryos at 24 and 48 hpf.

Subsection “Tenocyte projections are force sensors in the tendon ECM”, second paragraph: Due to technical issues with the FAC sorter and the weak fluorescence of our transgene at 24 hpf we were unable to FACS sort tenocytes for PCR. Therefore, we performed a real-time PCR analysis on 24 hpf and 48 hpf embryos (control and *αBtx* injected). We observed significant reductions in Tsp4b expression in *αBtx* samples only at 48 hpf. We include these results in Figure 3—figure supplement 3.

7) Depth-coding color scale is inconsistent, sometimes pink=0μm, sometimes blue=0μm. It is important to fix this because without a consistent scale one cannot judge changes in tenocyte projection length. Also, please adjust scale such that the soma of the tenocytes is always at the same depth/color to judge length defects.

Due to limitations in the software of the confocal microscope, we were unable to select for a range in the image stack to obtain uniform color coding. We have instead provided a scale bar that uses a uniform gradient bar where Blue is 0 and Red/Yellow is 95 μm.

8) Quantification of antibody fluorescence is tricky/error-prone. Ratio images between mCherry signal from scx:mCherry and pSMAD3 signal would be helpful to visualize changes. Ratio image of panels C divided by B and panels G divided by F would be informative in Figure 5 and 6.

In order to visualize the fine details of the projections, we have to overexpose the cell bodies in the red channel (mCherry). Hence, we normalized the green channel (pSMAD3 signal) with background fluorescence signal in the same channel. The background signal was obtained from a ROI that was similar in area to the nucleus but in an anuclear region. This was performed blindly by two authors (LK and AS) to avoid any user bias. The statistics were performed with these new normalized data points.

9) Force-dependent gene expression: αBTX looks convincing but SB431542-treated embryos show a mild reduction only. Nocodazole affects gene expression inconsistent with hypothesis/conclusion (scx goes up, ctgfa2 goes up – maybe because nocodazole just makes the embryos very sick?). If force induces TGFβ signaling which induces gene expression, then the changes in gene expression should be roughly equivalent in αBtx (no force) and SB431542-treatments (no TGFβ signaling). This is not the case, so force does not only act through TGFβ signaling (or the inhibitor does not fully inhibit – unlikely though since pSMAD3 is more reduced than in αBtx injections). Since qPCR is a global (and fairly noisy) assay, is there another way to look at gene expression (antibodies, reporter transgenes)? If so, scx:mCherry reporter/antibody embryos could be analyzed in the different conditions and mCh could serve as a reference/normalization (not in the case of nocodazole though).

While our results suggest that TGFβ signaling is a primary mechanotransduction pathway in tenocyte morphogenesis, they do not exclude other signaling pathways such as YAP/TAZ etc., which can also induce CTGF. Hence, reducing force has a stronger effect on gene expression because it shuts down all mechanotransduction pathways, while inhibitor treatments specifically affect TGFβ-dependent expression. To address this we made several attempts at isolation of mCherry positive tenocytes using FACS. The experiment involved dissociation of embryos to produce a cell suspension for sorting, which took about 45’ of enzymatic and mechanical dissociation. The process of sorting took another 2 hours as we had four samples to sort for each experiment. Hence, the tenocytes were dissociated from their native ECM environment for a prolonged period of time, which we believe must have stressed the cells. Hence, ddPCR and qPCR results from the extracted RNA showed random expression levels of genes (including housekeeping genes).

Subsection “Tenocyte projections regulate force-dependent gene expression”: Hence, we opted to perform a ddPCR reaction on RNA extracted from whole embryos, which showed absolute expression levels of genes that matched the pattern observed in our qPCR experiment. We have included these ddPCR data in Figure 6—figure supplement 2.

10) If αBTX injection nearly completely suppresses tsp4b mRNA expression by qPCR, why is there Tsp4b protein (Figure 3—figure supplement 1) in these embryos? The reduction in Tsp4b protein is by far not as striking. Could this be clarified? Also, is there a way to control for both initial levels of expression and the rate of decline of mRNA in different embryos? Is it possible that electrical stimulation not only results in restoring force but also signaling?

Muscle progenitors produce some Tsp4b prior to differentiation (see comment #6). Electrically stimulating muscle contractions in control embryos does not alter gene expression, suggesting that the changes we observe are due to mechanical force. As mentioned in the response to comment #6, we have performed a qPCR at an early embryonic stage – 24 hpf – and a later stage – 48 hpf – in both control and *αBtx* injected embryos. We observe a force-independent basal expression of Tsp4b at 24 hpf similar to the expression of Tsp4b in paralyzed 48 hpf embryos. From our previous work (Subramanian and Schilling, 2014) we know that Tsp4b is robustly expressed in the entire myotome until 30 hpf and that ectopic expression of Tsp4b RNA always produces protein that localizes to MTJs. Hence, we do not see a strong correlation between the protein level and RNA levels at later stages.

11) The authors contend that ECM organization of the vertical myosepta is under the control of tenocytes. However, they only look at one specialized component of the ECM, thrombospondin4. It would be helpful if Figure 3 could better address the issue of tenocytes' contribution to ECM organization as a whole. Although the authors have previously shown that Thrombospondin4 is an important regulator of muscle cell adhesion, it is unlikely to be the major component of the muscle extracellular matrix. The majority of the matrix is already secreted directly from the muscles themselves, and initial muscle attachment occurs in the absence of tenocytes. How do tenocytes reinforce or mature the ECM generally? Would it be feasible to stain for two or more major ECM components in the vertical myosepta, such as laminin, fibronectin or collagen? TEM could also be an excellent readout.

To measure effects of force on other tendon ECM proteins, we have performed immunohistochemical staining for fibronectin (Fn) and laminin (Lam) in paralyzed and stimulated embryos, which can be done in a few weeks. TEM would require 3-6 months minimum as UCI does not have a facility and we would rely on a core facility at UCSD (previously used to examine MTJ ultrastructure in Tsp4b-deficient or stimulated zebrafish). It is also unclear that TEM analyses would add critical additional information other than a higher resolution readout of ECM organization.

Subsection “Tenocyte projections are force sensors in the tendon ECM”, second paragraph: To analyze additional force dependent effects on MTJ ECM organization, we stained for Laminin and Fibronectin proteins that have been well studied in the context of MTJ development. We did not observe any significant changes in their localization pattern or fluorescence intensity at MTJs for either Lam or Fn in control versus *αBtx*-injected embryos. We have included a supplement figure showing the quantified data – Figure 3—figure supplement 1. Since, previous studies have shown that Fn is secreted from the developing myotome prior to muscle attachment and is later replaced by Lam between 24 hpf to 48 hpf, we hypothesize that these ECM components are not dependent on mechanical force from muscle contraction for their localization.

12) Image analysis/stats: The authors are lauded for attempting to quantify their data. However, the details provided do not necessarily provide confidence in how the data were analyzed.a) The image analysis section does not specifically address the issue of blinding, and certainly many of the measurements done depend tremendously on user input.b) How exactly were projections segmented and branch length was quantified? Was this done manually? If images were segmented in Fiji then the parameters of masking, etc. should be detailed.c) The t-test is appropriate for normal distributions, was this the case with the data obtained?d) The authors said power analyses were done but still 3 embryos per conditions seems quite low. Also, how were the 50 data points/embryo chosen? Were they always from the same anterior-posterior location? There could be a great deal of subjectivity in the choosing of those data points as well and theoretically they should have been randomly chosen.e) Given that the relevant details were not provided, it is likely that the analysis and stats were done in a less than ideal fashion. Rather than spend 2 months redoing them, one option would be for the authors to provide supplemental data with multiple image examples of each experiment.

1) We provide a detailed account of our statistical methods and quantification including the use of Image J plugins for branch point quantification, projection length measurements etc.

2) We have included detailed descriptions of all the statistical tests that were conducted and the conditions for choosing the appropriate tests. We have consulted multiple statisticians familiar with biological sampling to verify our statistical tests.

3) Wherever feasible we have included actual data points as dot plots to show variance within and across embryos in each sample.

4) Two authors, LK and AS, performed the measurements from image files with no knowledge of sample identity or condition. This helped limit user bias.

5) The projection length was measured manually using Simple Neurite Tracer. This plugin allows the user to map the projection across Z planes, computes the length of the projection and provides a visual representation of the track chosen by the software. We compared this track with the actual path of the project in a 3D reconstruction to confirm validity of the traces. Once confirmed the measurements were recorded for later analysis.

6) Before conducting a statistical test, we computed the variance and normality in the sample. For samples that lacked normality, we used Wilcoxon Rank Sum Test. For experiments where only two samples were compared, a t-test was only used when normality was established. For experiments involving more than 2 samples, we employed an ANOVA 1-way analysis with Tukey Post-Hoc test for individual comparisons, when equal variance was satisfied. In cases where equal variance was not satisfied, we used the Kruskal-Wallis test.

7) All images were captured at similar anatomical locations, between somites 16-19 in the embryos. We always quantified our data from the ventral VMS (MTJ) in these 4 somites. Datapoints were never controlled by the user. We collected all data from each tenocyte in the ventral VMS – projection length, fluorescence intensity, pSMAD3 fluorescence intensity. Hence, there is a variation in the number of datapoints between VMSs in an embryo and between embryos in the same sample.

We have provided these descriptions on our statistical methods under the Materials and methods section.